# DYNAMIC PARAMETRIC RETRIEVAL AUGMENTED GENERATION

## ABSTRACT

Retrieval-augmented generation (RAG) enhances large language models (LLMs) by injecting externally retrieved documents into the input context. It significantly increases inference costs and introduces knowledge conflicts, primarily caused by the lack of corresponding parametric knowledge in LLMs. Recently, Parametric RAG (PRAG) proposed to overcome these limitations by embedding symbolic documents into LLMs parameters, effectively reducing the inference costs and conflicts through offline training. However, PRAG needs to convert all documents into parameters in advance, which incurs high training and storage costs and renders it difficult to generalize to unseen documents. To address these challenges, we propose **Dynamic Parametric RAG (DyPRAG)**, a novel framework that leverages a lightweight parameter translator model to efficiently convert symbolic documents into parametric knowledge online. Specifically, the parameter translator employs several linear layers to convert document embeddings into LoRA modules of feed-forward networks of LLMs directly. DyPRAG achieves test-time parametric knowledge enhancement by dynamically generating the requisite parameters, which not only reduces the inference cost and mitigates knowledge conflicts inherent in RAG, but also lowers the training and storage overhead of PRAG. Extensive experiments on multiple datasets demonstrate the effectiveness and generalization capabilities of DyPRAG. Furthermore, the combination of contextual knowledge with test-time generated parametric knowledge offers a practical and more powerful RAG paradigm which updates parametric knowledge adaptively, enables superior knowledge fusion and alleviates knowledge conflicts in real-world applications. Our code is available at `https://anonymous.4open.science/r/DyPRAG_ICLR`.

## 1 INTRODUCTION

Large language models (LLMs) have demonstrated remarkable capabilities across diverse tasks (Liu et al., 2023), yet their performance on knowledge-intensive applications (Frisoni et al., 2024), such as question answering, remains constrained by limited access to up-to-date or domain-specific knowledge and a tendency to hallucination (Joshi et al., 2017; Kwiatkowski et al., 2019). To address this gap, retrieval-augmented generation (RAG) (Guu et al., 2020a) has emerged as a widely adopted approach which retrieves documents from external sources (e.g., Wikipedia) and injects them into the context, referred to as **in-context injection** (Izacard & Grave, 2021) (as shown in Figure 1 (a)).

While RAG mitigates knowledge gaps (Brown et al., 2020), this in-context injection strategy suffers from several limitations. As more documents are retrieved, inference costs increase rapidly due to elongated input sequences. More critically, knowledge conflicts (also known as RAG hallucination) often occur when external content contradicts the internal parametric knowledge of the LLM. It is mainly caused by low knowledge overlap, leading to erroneous outputs even in the presence of relevant documents(Zhang et al., 2024; Sun et al., 2024).

Parametric RAG (PRAG) employs another way of injecting knowledge, which integrates external knowledge directly into the parameters of LLMs to address these problems, known as **parameter injection** (as shown in Figure 1 (b)). The workflow of PRAG is divided into two stages. During the offline phase, PRAG first augments the retrieved documents to facilitate memorization and manipulation of knowledge (Allen-Zhu & Li, 2023a;b). Then, the augmented documents are fine-tuned with LoRA (Hu et al., 2022), encoding contextual knowledge directly into parameters. In the online

phase, retrieved documents are replaced with loadable parameters. Despite its benifit, PRAG faces critical limitations. Augmentation, training, and storing parameters for each retrieved document incur high computational and storage costs, coupled with non-generalization which struggles to adapt to unseen documents, severely limiting scalability in real-world applications (e.g., in frequent knowledge updates domains). This presents a critical challenge for PRAG: *How to achieve more efficient and generalizable test-time parametric knowledge enhancement with comparable performance?*

To address aforementioned challenges, we propose **Dynamic Parametric RAG (DyPRAG)**, a novel lightweight framework that enables on-the-fly parameter injection at test time (Figure 1 (c)). Rethinking PRAG, its intrinsic goal is to obtain an underlying mapping function $\mathcal{F}$ which transforms external documents into parameters by statically fine-tuning on each document. Instead of this, DyPRAG introduces the parameter translator $\mathcal{F}'_\phi$, a small hypernetwork trained offline to learn this generalized mapping. This model, once trained, can dynamically

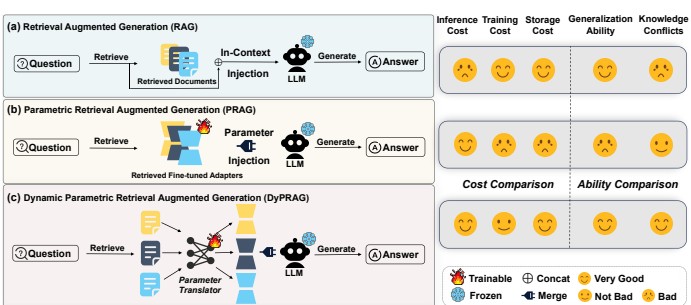

Figure 1: Compared to RAG and PRAG, the proposed DyPRAG generates parametric knowledge dynamically during test-time, which provides multiple benefits, including lower inference, training, and storage costs, better generalization, and mitigation of knowledge conflicts.

generate document-related parameters at inference time to enhance the parametric knowledge of LLMs. After a detailed analysis of computation and storage overhead, our method significantly reduces the high inference costs of traditional RAG while eliminating the rigid training and storage costs of PRAG.

Through extensive experiments, we derive two key findings: **1) DyPRAG enhances the test-time parametric knowledge of LLMs effectively.** During evaluation, DyPRAG outperforms standard RAG across different scales of LLMs, demonstrating its ability to enhance the internal knowledge of LLMs. Furthermore, although PRAG learns $\mathcal{F}$ by training separately for each document, DyPRAG achieves comparable or even better performance in various scenarios with significantly lower costs. **2) Combining test-time generated parametric knowledge with contextual knowledge leads to superior knowledge fusion.** Following Su et al. (2025), we further investigate the combination of in-context injection and parameter injection, referred to as **DyPRAG-Combine**. In both independent identically distributed and out-of-distribution settings, DyPRAG-Combine achieves the best results, showing strong generalization ability. Additionally, we find that DyPRAG-Combine effectively relieves knowledge conflicts by first injecting context-related parameters to fill the absent parametric knowledge gap. Based on the experimental results on RAGTruth (Niu et al., 2023) benchmark, we observe that DyPRAG-Combine enables LLMs to better internalize contextual knowledge, even on unseen documents. We further provide an in-depth analysis of how the parameter translator maps knowledge from different sources and how to interpret model performance based on internal signals. These findings suggest that integrating parametric and contextual knowledge using DyPRAG could be a promising approach for building a powerful and robust RAG system in real-world applications. We summarize our contributions as follows:

- We propose **Dynamic Parametric RAG (DyPRAG)**, a novel lightweight framework that efficiently converts symbolic documents into parameters at test-time. To the best of our knowledge, DyPRAG is the first approach in the RAG field to enable online transformation of symbolic knowledge into parametric representations, thereby eliminating the need for offline pre-conversion and storage of parameterized documents.

- We further develop a practical and powerful RAG paradigm **DyPRAG-Combine**, which effectively integrates symbolic documents with dynamically generated parametric knowledge, enabling the supplementation of requisite parametric knowledge in advance.

- Experimental results demonstrate that DyPRAG not only significantly outperforms in generalization but also efficiently enhances parametric knowledge and seamlessly integrates contextual knowledge,

boosting performance while reducing knowledge conflicts. As a result, DyPRAG provides a more powerful and robust RAG paradigm for real-world applications.

## 2 RELATED WORK

### 2.1 RETRIEVAL AUGMENTED GENERATION

Large language models (LLMs) have demonstrated remarkable performance across diverse applications. However, their inherent knowledge often falls short in handling knowledge-intensive tasks, highlighting the need for external knowledge integration to ensure robust performance in such contexts. A prominent approach to bridging this gap is retrieval-augmented generation (RAG), which augments LLMs by incorporating relevant external knowledge sources (Borgeaud et al., 2022; Wang et al., 2024a;b; Guu et al., 2020b). The retrieved documents are appended to the LLM's input context, enabling it to leverage knowledge beyond its training data (Lewis et al., 2020). However, this approach leads to high inference costs as the number and length of retrieved documents increase (Xiong et al., 2023). To address this issue, a recent study introduces Parametric RAG (PRAG) (Su et al., 2025), a paradigm that fine-tunes the model on augmented documents, encoding useful information into parameters. While PRAG mitigates the inference cost, it introduces additional training and storage costs due to the need to obtain and store LoRA parameters. Our proposed method significantly reduces the high costs associated with standard RAG and PRAG while achieving superior generalization. By combining contextual knowledge with test-time generated parametric knowledge via DyPRAG, our approach enables better knowledge fusion and effectively mitigating knowledge conflicts.

### 2.2 CONTEXT COMPRESSION

Context compression is widely adopted to improve the efficiency of LLMs in processing contextual knowledge. Recent studies propose condensing long contexts into soft prompts, allowing LLMs to utilize information more effectively (Mu et al., 2023; Ge et al., 2023). Meanwhile, other works focus on transforming context chunks into LoRA modules to improve the understanding ability of extended contexts (Mao et al., 2024; Wang et al., 2024c; Charakorn et al., 2025). xRAG (Cheng et al., 2024) integrates context compression by mapping documents into a compact token representation. Similarly, AAG (Liao et al., 2024b) draws inspiration from human cognition, retrieving and recalling relevant knowledge to compensate for knowledge gaps. This approach activates relevant information within LLMs without relying on external resources. Building upon these advancements, we present the first in-depth investigation into transforming symbolic documents into model parameters within RAG systems. Our study demonstrates that this approach effectively unifies the contextual and parametric knowledge, making it highly suitable for the RAG domain. This unification significantly mitigates knowledge conflicts and enhances overall performance in RAG systems.

## 3 METHODOLOGY

In this section, we introduce the Dynamic Parametric RAG framework, as shown in Figure 2. We first formulate the problem and review the previous PRAG framework. Specifically, we revisit the offline document parameterization process, which transforms documents into parametric representations through **Document Augmentation** and **Document Parameterizing**, following Su et al. (2025). Subsequently, we present our **Parameter Translation** process, which learns the underlying function to map document embeddings into feed-forward networks (FFN) parameters via LoRA (Hu et al., 2022). Once the translator is well optimized, retrieved documents can be directly converted into parametric representations online. These parameters can be efficiently integrated into LLMs, enhancing model parametric knowledge while reducing inference, training, and storage costs at test-time.

### 3.1 PRELIMINARY OF PARAMETRIC RAG

This subsection introduces the problem formulation of the RAG task and outlines the Parametric RAG pipeline proposed in Su et al. (2025).

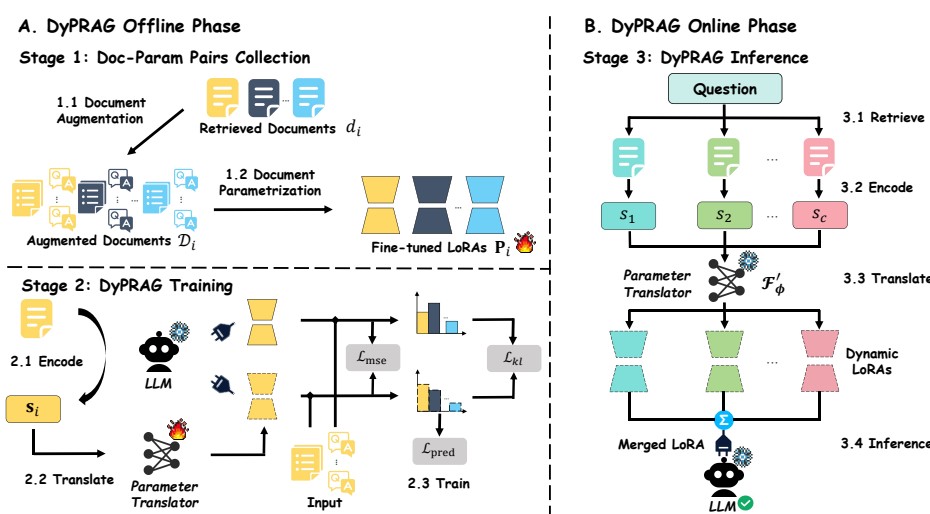

Figure 2: An illustration of the DyPRAG framework. The offline phase consists of two stages: **Stage 1** follows the same parameterization process as PRAG to collect document-parameter (Doc-Param) pairs. In **Stage 2**, a parameter translator $\mathcal{F}'_\phi$ is trained to learn a generalizable mapping from documents to corresponding parametric representations. During the online phase, **Stage 3** leverages the learned translator $\mathcal{F}'_\phi$ to dynamically generate LoRA modules for any document at test-time. This enables DyPRAG to enhance LLMs with external parametric knowledge on demand.

**Standard RAG.** Let $\mathcal{M}$ denote a large language model (LLM) with base parameters $\Theta$. Given a user query $q$, the task of LLM is to generate an accurate response augmented by an external corpus $\mathcal{C}$, expressed as $\mathcal{C} = \{d_1, d_2, \ldots, d_N\}$. Each element $d_i$, referred to as a document, represents a text chunk retrieved from external sources (Izacard & Grave, 2021). To achieve this, a retrieval module $\mathcal{R}$ is employed to compute relevance scores between $q$ and the documents in $\mathcal{C}$. Traditional RAG approaches select the top-$c$ documents with the highest similarity scores and concatenates them with the query to form the extended input context. Based on this augmented input, $\mathcal{M}$ generates the response by leveraging both the query and the retrieved documents. This procedure, referred to as **In-context Injection**, significantly increases the inference costs as the context length grows.

**Parametric RAG.** In contrast, Parametric RAG (PRAG) integrates documents directly into the parameters of $\mathcal{M}$ to reduce the cost associated with long contexts. Each document $d_i \in \mathcal{C}$ is transformed offline into a parametric representation $\mathbf{P}_i = \mathcal{F}(d_i)$, where $\mathcal{F}$ is an underlying mapping function that converts each document $d_i$ into its corresponding parameters $\mathbf{P}_i$. To achieve a more effective mapping, PRAG employs **Document Augmentation**, inspired by Allen-Zhu & Li (2023a;b), to help the model memorize and manipulate the information contained in document $d_i$. Specifically, PRAG uses $\mathcal{M}$ to rewrite $d_i$ into multiple variations, resulting in $\{d_i^1, d_i^2, \ldots, d_i^n\}$. Additionally, for each original document $d_i$, PRAG prompts $\mathcal{M}$ to generate $m$ question-answer (QA) pairs: $\{(q_i^1, a_i^1), (q_i^2, a_i^2), \ldots, (q_i^m, a_i^m)\}$, where $n$ and $m$ are hyperparameters. This augmented set of documents preserves the factual content of the original document while incorporating diverse linguistic variations, expressed as:

$$\mathcal{D}_i = \left\{ (d_i^k, q_i^j, a_i^j) \,\middle|\, k \in [1, n],\, j \in [1, m] \right\}, \tag{1}$$

where each triple $(d_i^k, q_i^j, a_i^j)$ is then concatenated to a training sample $x = [d_i^k \oplus q_i^j \oplus a_i^j]$. For **Document Parameterizing**, PRAG utilizes LoRA (Hu et al., 2022) to encode parametric knowledge

$\mathbf{P}_i$ for each $\mathcal{D}_i$ where the overall goal is to optimize:

$$\min_{\mathbf{P}_i} \sum_{(d_i{}^k, q_i{}^j, a_i{}^j) \in D_i} \sum_{t=1}^{T} -\log\ P_{\boldsymbol{\Theta}+\mathbf{P}_i}(x_t \mid x_{<t}), \tag{2}$$

where $\mathbf{P}_i$ is the trainable low-rank matrix and only apply to feed-forward network (FFN). During the inference phase, PRAG directly incorporates the obtained parametric representation $\mathbf{P}_i$ into the model parameters. We refer to this approach as **Parameter Injection**. Notably, although this method eliminates the use of documents as context, it further introduces significant training cost and storage cost, which will be analyzed in Section A.

## 3.2 DYNAMIC PARAMETRIC RAG

In this section, we describe the detailed process of our proposed **Dynamic Parametric RAG (DyPRAG)** paradigm. Rethinking PRAG, its intrinsic goal is to obtain a document-specific mapping function $\mathcal{F}$ through repeated augmentation and training for each document $d_i$ separately. However, this process is computationally intensive and impractical in real-world applications, where new documents require retraining from scratch. We argue that the key to optimization lies in addressing the question: *How to obtain a generalized mapping function $\mathcal{F}$?* To this end, we propose a three-stage framework designed to enable parameter injection in an effective and efficient manner, which eliminates the need to pre-convert and store parameters for all documents offline.

**Doc-Param Pairs Collection.** To derive the general mapping function $\mathcal{F}$, we start by collecting a set of document-parameter (Doc-Param) pairs using the method described in Sec. 3.1. For each document $d_i$, we collect its corresponding parametric representation $\mathbf{P}_i$, forming the alignment set $\mathcal{K} = \{(d_1, \mathbf{P}_1), (d_2, \mathbf{P}_2), \ldots, (d_N, \mathbf{P}_N)\}$.

**Dynamic Parametric RAG Training.** After obtaining the alignment set $\mathcal{K}$, we utilize the original LLM $\mathcal{M}$ to encode textual documents into embeddings. For a given document $d_i$, we extract the last hidden state $\mathbf{s}_i \in \mathbb{R}^h$ at the last token position before transforming it into the vocabulary space, where $h$ represents the hidden dimension. To model the implicit transformation, we design a simple hypernetwork called **Parameter Translator** $\mathcal{F}'_\phi$ to translate $\mathbf{s}_i$ into parametric representation $\mathbf{P}_i$. This hypernetwork consists of several linear layers parameterized by a base parameter $\phi$. As an example, consider the up-project module in FFN. The standard LoRA process as follows:

$$\mathbf{W}' = \mathbf{W} + \boldsymbol{\Delta}\mathbf{W} = \mathbf{W} + \mathbf{B}\mathbf{A} \tag{3}$$

where $\mathbf{W} \in \mathbb{R}^{h \times k}$, $\mathbf{B} \in \mathbb{R}^{h \times r}$ and $\mathbf{A} \in \mathbb{R}^{r \times k}$. $k$ represents the intermediate dimension of FFN and $r$ is the controllable LoRA rank. At training phase, $\mathcal{F}'_\phi$ performs separately on $\mathbf{B}$ and $\mathbf{A}$. Formally:

$$\mathbf{B}^l = \text{Reshape}(\mathbf{W}_{\text{up}}^{l,B} \text{Relu}(\mathbf{W}_{\text{down}}^{l,B}(\mathbf{s}_i \oplus \text{idx}^l))) \tag{4}$$

where $\mathbf{W}_{\text{down}}^{l,B} \in \mathbb{R}^{p \times (h+1)}$ and $\mathbf{W}_{\text{up}}^{l,B} \in \mathbb{R}^{hr \times p}$. Here, $p$ represents the tunable intermediate dimension of the MLP module in $\mathcal{F}'_\phi$, and Reshape($\cdot$) reshapes the output vector into the shape of $\mathbf{B}$. This process is applied at each layer $l$, so we concatenate the layer index with $\mathbf{s}_i$. We provide the visualization of this workflow in Appendix J. A similar procedure is followed for matrices $\mathbf{A}$ and in other modules of FFN. The parametric representation generated by $\mathcal{F}'_\phi$ is denoted as $\mathbf{P}'_i$. Our goal is for it to perform as effectively as $\mathbf{P}_i$.

To align with PRAG (Su et al., 2025), we utilize the augmented dataset $\mathcal{D}_i$ and the same objective function as presented in Eq. 2 to optimize $\mathcal{F}'_\phi$ which corresponds to $\mathcal{L}_{\text{pred}}$. Additionally, for the target LoRA adapter $\mathbf{P}_i$, we employ $\mathcal{L}_{\text{mse}}$ to compute the difference between the generated parameters and the target parameters. The Kullback-Leibler divergence (Polzehl & Spokoiny, 2006), denoted as $\mathcal{L}_{\text{kl}}$, quantifies the discrepancy in word probability distributions between the two models, with the model using $\mathbf{P}_i$ serving as the target distribution to be imitated. The overall formulation is given by:

$$\mathcal{L}_{\text{mse}} = \text{MSE}(\mathbf{P}_i, \mathcal{F}'_\phi(d_i)) \tag{5}$$

$$\mathcal{L}_{\text{kl}} = \text{KL}(P_{\boldsymbol{\Theta}+\mathbf{P}_i}(x \mid D_i), P_{\boldsymbol{\Theta}+\mathcal{F}'_\phi(d_i)}(x \mid D_i)) \tag{6}$$

$$\mathcal{L}_{\text{align}} = \mathcal{L}_{\text{pred}} + \lambda_1 \mathcal{L}_{\text{mse}} + \lambda_2 \mathcal{L}_{\text{kl}} \tag{7}$$

where we calculate the overall alignment loss for each document $d_i$, $\lambda_1$ and $\lambda_2$ are tunable hyperparameter which set to 100 and 0.01 separately to make loss range similar.

**Dynamic Parametric RAG Inference.**   During the inference stage, once a well-trained parameter translator $\mathcal{F}'_\phi$ is obtained, we can efficiently perform parameter injection which significantly reduces the inference costs. For a test query $q^t$, we rerun the retrieval process using the retrieval module $\mathcal{R}$ to select the most relevant documents. For each selected document $d_i^t$, we derive its embedding $\mathbf{s}_i^t$ and input it into $\mathcal{F}'_\phi$ to obtain the dynamic LoRA adapter $\mathbf{P}_i^{t,'}$, which encodes the relevant information from the document in parameter. We then merge this as the LoRA parameter for inference, resulting in low inference costs without requiring the concatenated documents.

# 4   EXPERIMENTS

## 4.1   EXPERIMENTS DETAILS

**Datasets.**   We validate our approach using various benchmarks to evaluate distinct reasoning abilities, including multi-hop reasoning and commonsense inference. The selected datasets are **2WikiMultihopQA (2WQA)** (Ho et al., 2020), **HotpotQA (HQA)** (Yang et al., 2018), **PopQA (PQA)** (Mallen et al., 2022) and **ComplexWebQuestions (CWQ)** (Talmor & Berant, 2018). We provide detailed information about these datasets in Appendix B.1.

**Evaluation Metrics.**   For evaluation, we use the Exact Match (EM) score (%) to compare the extracted answer with the reference answer at the exact match level. Additionally, we employ the F1 score (%), which balances precision and recall by considering partially correct answers. Both 2WQA and HQA categorize questions by reasoning type, with 2WQA having four categories and HQA two. To compare DyPRAG with other RAG baselines across reasoning tasks, we use the first 300 questions from each sub-dataset for evaluation.

**Implementation Details.**   To ensure broad effectiveness across models, we select LLMs of varying scales and series, including Qwen2.5-1.5B-Instruct (Yang et al., 2024), LLaMA-3.2-1B-Instruct (Meta, 2024b) and LLaMA-3-8B-Instruct (Meta, 2024a). For our base experiments, we collect 200 additional questions from each non-overlapping sub-dataset. The number of retrieved documents $c$ is set to 3, resulting in a alignment set $\mathcal{K}$ of 4,800 samples. The intermediate size $p$ is set to 32. All experiments were conducted using PyTorch on NVIDIA A100 GPUs (80GB). Please refer to Appendix B.1 for more detailed settings.

## 4.2   BASELINES

We select the following baselines to compare with our proposed **DyPRAG**, detailed in Appendix B.1:

- **Vanilla** represents the answer from original LLMs without any external knowledge.
- **RAG** appends top-retrieved documents to the LLM's input prompt, explicitly instructing the model to reference them when answering.
- **PRAG** injects relevant documents into the LLM's parameters via offline parameterization, reducing reliance on retrieved documents.
- **SFT** fine-tunes LLMs with same setting in DyPRAG to encode all knowledge without context.
- **Context-DPO** (Bi et al., 2024) aligns LLMs through direct preference optimization (DPO) (Rafailov et al., 2023) to enhance context-faithfulness of LLMs and inference with retrieved documents.

Following the approach in Su et al. (2025), we conduct experiments that combine both in-context and parameter injection to explore their interaction. Specifically, the retrieved documents are appended to the input context, and their corresponding parametric representations are integrated into the model. This results in two additional baselines, referred to as **PRAG-Combine** and **DyPRAG-Combine**.

## 4.3   MAIN RESULTS

In this section, we present the main experimental results and a detailed analysis of DyPRAG in comparison with the selected baselines. Additionally, we provide efficient RAG baselines in Appendix C. Notably, the vanilla model occasionally outperforms RAG in certain situations. We analyze the reasons for this in Appendix H and confirm that it won't affect the subsequent analysis.

Table 1: The experimental results of DyPRAG are compared with parametric RAG, standard RAG and two training-based methods. All metrics are reported as EM scores (%) and F1 scores (%). The best performance is bolded, while the second-best is underlined. The **Avg** is the average performance over all tasks.

| Base LLM | Method | 2WQA | | HQA | | PQA | | CWQ | | Avg | |
|---|---|---|---|---|---|---|---|---|---|---|---|
| | | EM | F1 | EM | F1 | EM | F1 | EM | F1 | EM | F1 |
| LLaMA3.2-1B | Vanilla | 17.47 | 22.87 | 18.56 | 24.10 | 0.67 | 2.26 | 23.67 | 34.94 | 16.74 | 21.04 |
| | SFT | 8.67 | 11.25 | 1.67 | 2.96 | 0.00 | 1.33 | 7.67 | 12.77 | 5.60 | 7.92 |
| | Context-DPO | 19.33 | 24.14 | 17.00 | 23.35 | 4.00 | 12.79 | 7.67 | 13.00 | 15.93 | 21.66 |
| | RAG | 17.93 | 24.77 | 21.44 | 30.33 | 9.67 | 17.65 | 25.67 | 37.39 | 18.93 | 26.99 |
| | PRAG | 20.13 | 25.92 | 19.00 | 25.35 | 12.00 | 23.58 | 26.00 | 35.86 | 19.57 | 26.51 |
| | PRAG-Combine | 20.60 | 26.94 | 23.33 | 30.81 | 20.33 | 31.07 | 28.33 | 39.63 | 22.17 | 29.78 |
| | DyPRAG (ours) | 24.27 | 29.91 | 19.56 | 25.97 | 7.33 | 11.33 | 28.33 | 36.86 | 21.57 | 27.57 |
| | DyPRAG-Combine (ours) | 26.33 | 32.53 | 23.33 | 30.80 | 21.33 | 29.93 | 29.33 | 38.96 | 25.23 | 31.80 |
| Qwen2.5-1.5B | Vanilla | 20.87 | 27.20 | 14.78 | 23.13 | 0.67 | 2.87 | 18.00 | 26.47 | 16.74 | 25.79 |
| | SFT | 18.60 | 22.61 | 8.78 | 13.63 | 0.00 | 6.95 | 4.67 | 13.96 | 12.40 | 17.49 |
| | Context-DPO | 17.60 | 24.35 | 15.00 | 24.35 | 0.33 | 14.18 | 12.33 | 19.20 | 14.57 | 22.82 |
| | RAG | 16.33 | 23.89 | 14.89 | 24.68 | 0.67 | 9.97 | 18.64 | 28.23 | 14.56 | 23.17 |
| | PRAG | 21.93 | 29.38 | 16.00 | 24.04 | 1.33 | 3.87 | 22.31 | 30.82 | 18.13 | 25.37 |
| | PRAG-Combine | 19.07 | 27.29 | 19.33 | 26.15 | 2.67 | 12.61 | 21.67 | 32.13 | 17.77 | 25.96 |
| | DyPRAG (ours) | 21.87 | 28.46 | 17.11 | 24.93 | 3.00 | 6.64 | 22.67 | 31.94 | 18.64 | 25.56 |
| | DyPRAG-Combine (ours) | 18.87 | 25.87 | 20.67 | 30.13 | 7.33 | 22.69 | 23.67 | 33.57 | 18.74 | 27.60 |
| LLaMA3-8B | Vanilla | 30.00 | 36.43 | 19.89 | 28.64 | 4.67 | 7.96 | 30.00 | 42.44 | 24.43 | 31.85 |
| | SFT | 1.53 | 13.09 | 0.33 | 2.19 | 0.00 | 0.00 | 0.00 | 5.92 | 0.86 | 7.80 |
| | Context-DPO | 14.93 | 24.42 | 12.45 | 21.67 | 4.33 | 18.68 | 8.00 | 13.81 | 12.43 | 21.96 |
| | RAG | 28.40 | 34.20 | 19.13 | 28.67 | 5.67 | 16.13 | 25.33 | 35.45 | 23.04 | 30.86 |
| | PRAG | 33.20 | 40.54 | 35.55 | 45.88 | 20.33 | 26.13 | 32.67 | 43.54 | 32.57 | 41.00 |
| | PRAG-Combine | 34.47 | 42.20 | 40.11 | 50.82 | 11.33 | 26.23 | 28.00 | 36.41 | 33.20 | 42.61 |
| | DyPRAG (ours) | 32.07 | 39.17 | 24.67 | 37.33 | 11.00 | 13.60 | 32.67 | 41.87 | 27.80 | 36.23 |
| | DyPRAG-Combine (ours) | 36.33 | 47.68 | 33.22 | 43.22 | 21.00 | 32.86 | 29.67 | 39.07 | 33.20 | 43.69 |

**Overall Analysis.** Since PRAG learns the mapping function $\mathcal{F}$ by training separately for each document, it can be considered as a upper bound of DyPRAG. Remarkably, our proposed DyPRAG achieves comparable or even superior results across various tasks, as shown in Table 1. For instance, using LLaMA3.2-1B, DyPRAG achieves an average score of 27.57% (21.57%), surpassing PRAG by 1.06% (2.00%), RAG by 0.58% (2.64%) and vanilla by 5.18% (4.83%) in F1 (EM) scores. This demonstrates that our method learns more useful information when trained on diverse datasets. We also compare DyPRAG with efficient RAG baselines, including FLARE (Jiang et al., 2023) and DRA-GIN (Su et al., 2024). As shown in Table 8, both DRAGIN and FLARE outperform standard RAG in most 2WQA settings. However, DyPRAG achieves even better results, demonstrating its superiority. For example, when using LLaMA3-8B as the base model, DyPRAG outperforms DRAGIN and FLARE by 1.56% and 2.63% on 2WQA in F1 scores, respectively. However, Context-DPO proves less effective in resolving knowledge conflicts, while SFT experiences severe collapse, failing to encode such a large amount of knowledge, which leads to significant performance degradation. These results highlight the consistent performance improvements offered by DyPRAG over all baselines, underscoring its effectiveness for test-time parametric knowledge enhancement.

**DyPRAG-Combine Leads to Superior Performance.** By combining in-context injection with parameter injection, DyPRAG-Combine achieves the best performance across all models, outperforming all baselines. For instance, DyPRAG-Combine outperforms PRAG-Combine by 2.02% (3.06%) on LLaMA3.2-1B, 0.55% (0.17%) on Qwen2.5-1.5B and 1.08% (0.00%) on LLaMA3-8B on average in F1 (EM) scores. Moreover, combining these two types of knowledge results in shorter responses, effectively reducing costs due to improved knowledge internalization ability, as shown in Figure 4. These results demonstrate the dynamic parameters generated by our approach effectively intergrade with contextual knowledge, enabling these two information sources to complement each other.

## 4.4 OUT-OF-DISTRIBUTION PERFORMANCE

To further demonstrate the generalization ability of the DyPRAG method, we evaluate it in the out-of-distribution (OOD) scenario. Notably, PRAG can not handle this OOD scenario without additional offline training. We conduct the OOD performance on commonsense datasets: StrategyQA (**SQA**)(Geva et al., 2021), **IIRC**(Ferguson et al., 2020), and OpenBookQA (**OBQA**)(Mihaylov et al., 2018). Additionally, MedMCQA (**MQA**)(Pal et al., 2022) focuses on a completely unseen domain about medical. All OOD datasets are provided with ground-truth passages.

As shown in Table 2, the vanilla model performs poorly due to a lack of sufficient relevant knowledge, particularly in the IIRC dataset. DyPRAG effectively enhances parametric knowledge, resulting

Table 2: The OOD performance on three open-domain datasets for $\mathcal{F}'_\phi$ trained on $\mathcal{K}$ is reported.

| Base Model | Method | IIRC | SQA | OBQA | MedQA | Avg |
|---|---|---|---|---|---|---|
| LLaMA3.2-1B | Vanilla | 10.99 | 21.67 | 40.33 | 39.00 | 28.00 |
| | SFT | 2.83 | 0.00 | 0.00 | 0.00 | 0.71 |
| | RAG | 40.38 | 27.67 | **52.00** | 50.33 | 42.60 |
| | DyPRAG | 14.04 | 39.67 | 43.00 | 40.67 | 34.35 |
| | DyPRAG-Combine | **41.91** | **50.33** | **52.00** | **52.67** | **49.23** |
| Qwen2.5-1.5B | Vanilla | 8.78 | 1.00 | 40.09 | 33.67 | 20.89 |
| | SFT | 7.39 | 0.00 | 9.67 | 0.00 | 4.27 |
| | RAG | 30.52 | 39.00 | 45.00 | **52.67** | 41.80 |
| | DyPRAG | 10.23 | 15.67 | 43.38 | 34.67 | 25.99 |
| | DyPRAG-Combine | **38.25** | **43.33** | **48.57** | **52.67** | **45.71** |
| LLaMA3-8B | Vanilla | 13.23 | 33.33 | 52.33 | 55.00 | 38.47 |
| | SFT | 2.42 | 0.00 | 0.00 | 0.00 | 0.61 |
| | RAG | 43.27 | 45.67 | 60.00 | 55.67 | 51.15 |
| | DyPRAG | 18.16 | 45.67 | 53.00 | 55.00 | 42.96 |
| | DyPRAG-Combine | **57.90** | **58.67** | **60.67** | 56.67 | **58.48** |

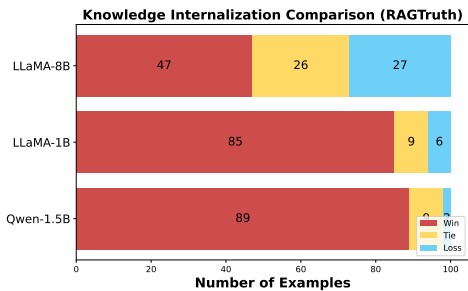

Figure 3: Comparison of knowledge internalization between DyPRAG-Combine vs RAG judged by GPT-4o.

Table 3: Ablation study of alignment loss. The backbone model is the LLaMA3.2-1B.

| Method | 2WQA | | | | | HQA | | | PQA | CWQ | Avg |
|---|---|---|---|---|---|---|---|---|---|---|---|
| | Compare | Bridge | Inference | Compose | Total | Bridge | Compare | Total | | | |
| **DyPRAG** | 51.25 | **48.15** | **17.35** | 7.54 | 25.31 | **14.05** | 43.9 | 19.97 | 8.37 | 36.86 | 25.28 |
| *Ablation Study* | | | | | | | | | | | |
| w/o $\mathcal{L}_{kl}$ | 29.54 | 37.74 | 12.94 | 5.78 | 20.27 | 7.35 | 35.28 | 13.12 | 1.93 | 22.85 | 18.68 |
| w/o $\mathcal{L}_{mse}$ | **56.06** | 36.96 | 17.29 | **8.40** | **27.28** | 12.78 | 42.11 | 17.65 | 5.94 | 32.98 | 23.38 |
| w/o $\mathcal{L}_{kl}, \mathcal{L}_{mse}$ | 45.23 | 24.84 | 16.74 | 7.48 | 23.43 | 12.66 | 39.46 | 18.26 | 2.42 | 34.92 | 22.54 |

in a moderate improvement in performance. However, when the model heavily relies on critical information from documents to answer questions, DyPRAG struggles to accurately reconstruct this information. This loss of information primarily stems from the encoding and translation processes, which contributes to the model's suboptimal performance (e.g., in IIRC). Notably, DyPRAG-Combine which incorporates golden passages with document-related parametric knowledge leads to deeper knowledge fusion, achieving best performance across all scenarios, even in hardest IIRC task. For example, DyPRAG-Combine improves performance on SQA (MQA) by 22.66% (2.34%) using LLaMA3.2-1B, on IIRC by 13.63% using LLaMA3-8B and on OBQA by 3.57% using Qwen2.5-1.5B. We believe that the observed performance gain comes from the coarse-grained parametric transformation of document knowledge. This transformation increases the overlap between the two distinct types of knowledge, thereby improving the model's ability to understand unseen documents. We further investigate DyPRAG's performance on non-QA tasks in **Appendix C**. Additionally, **Appendix F** presents textual similarity analyses across datasets to validate a reliable OOD setting, while **Appendix G** examines the generated LoRA matrices to explore the parameter translator's underlying generalization ability.

## 4.5 Ablation Study

**Effect of Alignment Loss.** The alignment loss $\mathcal{L}_{align}$ is composed of three components: $\mathcal{L}_{pred}$, $\mathcal{L}_{mse}$, and $\mathcal{L}_{kl}$. We investigated which component contributes the most to the effectiveness of DyPRAG. As shown in Table 3, removing any single loss component negatively impacts the model's performance. For instance, when $\mathcal{L}_{kl}$ is removed, the model's performance drops significantly, demonstrating that aligning with the target output distribution is an effective strategy (Liao et al., 2024a). While removing $\mathcal{L}_{mse}$ has the smallest impact, ensuring that $\mathcal{F}'_\phi$ generates $\mathbf{P}'$ values as close as possible to the trained $\mathbf{P}$ still proves beneficial. Furthermore, even when only $\mathcal{L}_{pred}$ is retained, DyPRAG maintains stable performance, indicating that the $\mathcal{L}_{pred}$ loss plays a central role in the overall alignment. We further present several ablation studies, including the effects of training dataset size, injected documents number to performance, intermediate dimension $p$, different retrievers, and data augmentation, as detailed in Appendix D.

**Effect of Data Augmentation.** In Section 3, we introduce data augmentation to improve the model's ability to memorize and process information from documents. To assess the impact of data augmentation on the DyPRAG method, we remove it during the Doc-Param pair collection phase and compare the results with those of the original method. The results in Table 4 indicate that removing

Table 4: Ablation study of effectiveness in data augmentation. All metrics are reported as F1 scores (%). The backbone model is the Qwen2.5-1.5B.

| Method | 2WQA (Total) | HQA (Total) | PQA | CWQ | IIRC | SQA | OBQA | MQA |
|---|---|---|---|---|---|---|---|---|
| **Vanilla** | 26.87 | 17.76 | 2.87 | 26.47 | 8.78 | 1.00 | 40.09 | 33.67 |
| **RAG** | 24.31 | 20.73 | 9.97 | 28.23 | 30.52 | 39.00 | 45.00 | 52.67 |
| **PRAG-Combine** | 27.49 | 23.10 | 23.43 | 32.13 | – | – | – | – |
| w/o Aug | 22.79 | 19.00 | 10.74 | 28.54 | – | – | – | – |
| *Change* | -17.1% | -17.7% | -54.2% | -11.2% | – | – | – | – |
| **DyPRAG** | 26.46 | 19.67 | 6.64 | 31.94 | 10.23 | 15.67 | 43.38 | 34.67 |
| w/o Aug | 28.36 | 15.71 | 3.35 | 28.04 | 8.49 | 0.30 | 38.36 | 22.94 |
| *Change* | +7.2% | -20.1% | -49.5% | -12.2% | -17.0% | -98.1% | -11.6% | -33.8% |
| **DyPRAG-Combine** | 25.18 | 27.57 | 22.69 | 33.57 | 38.25 | 43.33 | 48.57 | 52.67 |
| w/o Aug | 23.00 | 19.88 | 9.84 | 27.97 | 29.41 | 30.67 | 43.90 | 34.03 |
| *Change* | -8.7% | -27.9% | -56.6% | -16.7% | -23.1% | -29.2% | -9.6% | -35.4% |

data augmentation greatly diminishes the quality of offline parameterization, which in turn affects the parameter translator's ability to convert documents into parametric knowledge. This degradation results in a significant performance drop for both PRAG, which relies on offline parameterization, and DyPRAG, which dynamically converts parameters.

**Effect of Parameter Translators Size.** As illustrated in Section A, the total storage cost for $\mathcal{F}'_\phi$ is $3L(phr + 2p(h + 1) + pkr)$, which scales linearly with $p$. Therefore, we conducted an ablation study on $p$. As shown in Table 11, DyPRAG consistently outperforms both standard RAG and PRAG. Surprisingly, $p = 2$ achieves the second-best performance with a storage cost of only 7.71MB. In contrast, PRAG requires 9.33GB to store data for all test questions in main experiments, resulting in a significant overhead. The experiments demonstrate that our proposed DyPRAG not only drastically reduces storage costs but also enhances performance, showcasing exceptional robustness. Notably, DyPRAG significantly reduces the inference cost compared to RAG, while introducing only minimal overhead (i.e., encode and translate processes). We further present an end-to-end latency analysis in Table 12, demonstrating that both DyPRAG and DyPRAG-Combine achieve faster inference. This improvement is attributed to shorter responses and the asynchronous mode, which mitigates the latency introduced by the encoding and translation processes in real-world scenarios.

## 4.6 ANALYSIS OF CONTEXTUAL AND PARAMETRIC KNOWLEDGE CONFLICTS AND FUSION

**Pre-inject Converted Parameters Enhances Knowledge Overlap.** When LLMs struggle to identify the more reliable information source (Tao et al., 2024; Zhang et al., 2024), it is primarily due to conflicts between contextual knowledge and parametric knowledge, which fundamentally stem from low overlap between these two types of knowledge. We further investigate how internal signals, such as entropy, can detect RAG conflicts and how dynamic parametrization effectively mitigates these issues. As shown in Table 5, we leverage Entropy (EN), Length Normalized Entropy (LEN) (Malinin & Gales, 2020), and Lexical Similarity (LS) (Lin et al., 2022) to evaluate the likelihood of knowledge conflicts. Our findings indicate that EN and LEN increase, while LS decreases, when in-context injection is applied, suggesting that retrieved passages in RAG systems often exhibit low overlap with the model's internal knowledge that increasing the uncertainty during generation. Notably, a comparison between RAG and DyPRAG-Combine shows that employing the parameter translator to inject converted parametric knowledge significantly reduces knowledge conflicts, underscoring the effectiveness of DyPRAG.

In contrast, DyPRAG-Combine effectively integrates contextual knowledge with transformed parametric knowledge, enabling it to provide correct answers and demonstrating its ability to leverage both types of knowledge effectively. Compared to RAG, DyPRAG-Combine transforms the retrieved documents into parameters before concatenated into the input prompt. This approach ensures that the LLM already contains relevant knowledge when answering the questions, mitigating the well-known conflicts issues (Sun et al., 2024).

Table 5: We present the experimental results for the knowledge conflicts metrics of DyPRAG and DyPRAG-Combine, in comparison with Vanilla and Standard RAG. In these metrics, ↑ indicates that higher values are better, while ↓ indicates the opposite. The best performance for each metric is highlighted in bold. The backbone model is the LLaMA3.2-1B.

| Metric | Method | 2WQA (total) | HQA (total) | PQA | CWQ | SQA | IIRC |
|---|---|---|---|---|---|---|---|
| EN ↓ | Vanilla | 3.187 | 3.176 | 3.251 | 3.163 | 3.178 | 3.011 |
| | DyPRAG (ours) | **2.199** | **1.999** | **1.757** | **2.860** | **2.805** | **2.544** |
| | RAG | 3.565 | 3.453 | 3.778 | 3.619 | 3.398 | 3.030 |
| | DyPRAG-Combine (ours) | **2.755** | **2.470** | 3.584 | 3.467 | 3.136 | 2.555 |
| LEN ↓ | Vanilla | 0.637 | 0.635 | 0.650 | 0.633 | 0.636 | 0.602 |
| | DyPRAG (ours) | **0.440** | **0.400** | **0.586** | **0.572** | **0.561** | **0.509** |
| | RAG | 0.713 | 0.691 | 0.756 | 0.724 | 0.680 | 0.606 |
| | DyPRAG-Combine (ours) | **0.551** | **0.494** | 0.719 | 0.693 | 0.627 | 0.511 |
| LS ↑ | Vanilla | **0.923** | 0.936 | 0.723 | 0.730 | 0.497 | 0.963 |
| | DyPRAG (ours) | 0.915 | 0.933 | **0.842** | **0.859** | 0.527 | **0.966** |
| | RAG | 0.945 | 0.956 | 0.936 | 0.962 | 0.812 | 0.966 |
| | DyPRAG-Combine (ours) | **0.953** | **0.959** | **0.966** | **0.988** | **0.853** | **0.975** |

**DyPRAG Enables LLMs to Internalize Unseen Knowledge.** The retrieved documents in our experiments are primarily sourced from Wikipedia, which are already encountered by LLMs during pre-training. In this section, we further investigate how DyPRAG performs on unseen documents using the RAGTruth benchmark (Niu et al., 2023). Specifically, we randomly sample 100 examples from the QA-type sub-dataset, which presents greater challenges (e.g., the required answers are only accessible in carefully crafted context). As shown in Figure 3, DyPRAG-Combine significantly outperforms RAG. This demonstrates that DyPRAG effectively enables LLMs to better internalize contextual knowledge and mitigate conflicts, even when handling unseen data. Additionally, we present a further detailed analysis of contextual and parametric knowledge in **Appendix I**.

Table 6: Case study about contextual and parametric knowledge conflicts in 2WQA where only DyPRAG-Combine answers correctly (11.33%). The backbone model is the LLaMA3.2-1B. ▨ (red): deficiency in parametric knowledge, ▨ (yellow): knowledge conflicts, ▨ (green): successful knowledge manipulation.

| | |
|---|---|
| **Question:** Which film whose director was born first, The Snake Brothers or Olympus Has Fallen ? | |
| **Ground truth:** Behind Prison Gates | |
| **Retrieved top-1 document:** Roman Waugh was announced as director for the film. Olympus Has Fallen (film series)... | |

| Method | Answer | Status |
|---|---|---|
| **Vanilla** | David R | ✗ |
| **RAG** | The Snake Brothers | ✗ |
| **DyPRAG** (ours) | The Snake Brothers | ✗ |
| **DyPRAG-Combine** (ours) | Olympus Has Fallen | ✓ |

These experiments further validate our explanation in Section 4.4, demonstrating that DyPRAG struggles with fine-grained information reconstruction due to the inherent constraints of compression. However, the current results are sufficient to show that injecting transformed parametric knowledge increases its overlap with contextual knowledge while effectively mitigating the issue of uncertain responses caused by knowledge conflicts.

## 5 CONCLUSION

In this work, we propose Dynamic Parametric RAG (DyPRAG), a novel framework that addresses the high inference cost of RAG, the high training and storage costs of parametric RAG, while effectively mitigating knowledge conflicts. DyPRAG successfully learns the underlying mapping function from documents to parameters by leveraging a hypernetwork, enabling effective parametric knowledge enhancement at test-time. Extensive experiments conducted on multiple datasets demonstrate the superior performance, flexibility, and practicality of DyPRAG. By dynamically combining test-time generated parametric knowledge with contextual knowledge, DyPRAG enables adaptive parametric knowledge updates, superior knowledge fusion, and effective mitigation of knowledge conflicts. These advantages establish DyPRAG as a powerful and cost-efficient framework, highlighting its potential for real-world RAG applications.

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

Table 7: Comparison of cost metrics for different baselines. ATTN denotes the time complexity of the self-attention module as $O(|I|^2 h)$, and FFN represents the FFN with $O(|I|h^2)$, where context length $|I| = 1$ and $|R|$ denotes the response length. ▮ indicates significantly high cost, __ denotes negligible cost, and ╱ represents temporal storage.

| Method | Inference Cost | Training Cost | Storage Cost |
|---|---|---|---|
| RAG | $|R| \times ((c|d| + |q|)^2 \times \text{ATTN} + (c|d| + |q|) \times \text{FFN})$ | - | - |
| PRAG | $|R| \times (|q|^2 \times \text{ATTN} + |q| \times \text{FFN})$ | $M \times (9|d|^2 \times \text{ATTN} + 3|d| \times \text{FFN}) +$ $M \times E_1 \times (81|d|^2 \times \text{ATTN} + 9|d| \times \text{FFN})$ | $M \times 3Lr(h + k)$ |
| DyPRAG | $c \times (|d|^2 \times \text{ATTN} + |d| \times \text{FFN}) +$ $c \times O(p(h+1+hr)) +$ $|R| \times (|q|^2 \times \text{ATTN} + |q| \times \text{FFN})$ | $N \times (9|d|^2 \times \text{ATTN} + 3|d| \times \text{FFN}) +$ $N \times E_1 \times (81|d|^2 \times \text{ATTN} + 9|d| * \text{FFN}) +$ $N \times E_2 \times (9(|qa| + |d|)^2 \times \text{ATTN} + 3(|qa| + |d|) \times \text{FFN}) + O(p(h+1+hr))$ | $\cancel{N * 3Lr(h+k)} +$ $3L(phr + 2p(h+1) + pkr)$ |

# A    COMPUTATION AND STORAGE COST ANALYSIS

We present an initial pilot analysis and a broad evaluation of computation and storage costs across three baseline methods. More detailed analysis of time complexity is provided in Appendix A.1.

**Computation Cost.**    The computation cost in RAG is primarily the inference cost, whereas PRAG introduces additional training and inference costs due to augmentation and offline training. Suppose the average token count of document $d$ is $|d|$. As noted in Su et al. (2025), the augmentation process typically generates about $2|d|$ tokens, leading to an augmentation cost of $3|d|$. When training the target LoRA, a forward pass over $3|d|$ tokens and a backward pass over $6|d|$ tokens (typically twice the forward cost) result in a total training cost of $9|d|$. Although these tasks can be performed offline, it still requires a long time and do not generalize to new questions with unseen documents. In contrast, DyPRAG offers a more practical solution by requiring only $N$ Doc-Param pairs while even a small $N$ can achieve powerful performance, significantly reducing costs for augmentation and training. The cost of MLP-based $\mathcal{F}'_\phi$ is negligible compared to transformer-based LLMs (Vaswani et al., 2017).

The primary advantage of PRAG is the reduction of inference cost. Let $|q|$ denotes the length of the question, $c$ represents the number of retrieved documents. The inference context of PRAG and DyPRAG is $|q|$, whereas RAG requires $c|d| + |q|$. The parameterization process significantly reduces the inference cost, especially when $|d|$ and $c$ grow larger. Notably, the inference cost is also closely tied to the length of model response. DyPRAG demonstrates an improved ability to internalize knowledge, resulting in shorter responses that effectively reduce costs, as shown in Figure 4.

**Storage Cost.**    One of the main shortcomings of PRAG is the storage cost associated with $\mathbf{P}_i$. Let $r$ denote the LoRA rank, $L$ the number of Transformer layers, $h$ the hidden size, and $k$ the intermediate size of the FFN. The number of parameters in the parametric representation of a document is $3Lr(h + k)$. For instance, in the Qwen2.5-1.5B model (which has 28 layers, a hidden dimension of 1536, and an intermediate size of 8960), setting $r$ to 2 results in approximately 1.76M parameters, storing 3.36MB in 16-bit precision for each parametric representation. In our following experiments, we need to store 9.33GB offline parameters for Qwen2.5-1.5B, presenting a significant storage cost.

In contrast, our DyPRAG only needs to save the weights of $\mathcal{F}'_\phi$. As we set the intermediate size $p$ of the $\mathcal{F}'_\phi$ to 2, the total number of parameters for the Qwen2.5-1.5B model is $3L(phr + 2p(h+1) + pkr)$ as we configure separate translators for up-proj, down-proj, and gate-proj. This amounts to about 4.04M parameters, storing only 7.71MB (0.08% of PRAG) in 16-bit precision. The reduced storage cost makes it negligible compared to its generalization ability when used in real applications.

## A.1    DETAILED COST COMPARISON

In this section, we provide a detail comparison of several cost metrics for standard RAG, PRAG and our proposed DyPRAG, as shown in Table 7.

**Inference Cost.**    We first analyze the inference cost across three baselines. Intuitively, the RAG method requires more resources for inference due to its context length of $c|d| + |q|$, compared to only $|q|$ for PRAG and DyPRAG. In our experimental settings, $|q|$ is usually less than 100, while $|d|$ is typically larger than 600, with $c$ set to 3. This results in an attention cost of at least 271x and a FFN cost of 19x for RAG. For DyPRAG, there is additional cost incurred for encoding and

translating. The encoding cost is $c \times (|d|^2 \times \text{ATTN} + |d| \times \text{FFN})$, as each document should be encoded separately. As shown in Table 11, the encoding time is significantly lower than the inference time because encoding requires only a single forward pass. Additionally, the translation time is also negligible. Moreover, the response length $|R|$ exhibits a linear relationship with the LLM inference loss. As illustrated in Figure 4, the response length decreases when DyPRAG is employed, enabling LLMs to better internalize knowledge. Notably, DyPRAG-Combine achieves much shorter response lengths, significantly reducing inference costs compared to standard RAG.

**Training Cost.** PRAG (Su et al., 2025) introduces further training for each document to obtain corresponding LoRA parameters. In Section A, we hypothesize that after augmentation, there are a total of $3|d|$ tokens, resulting in a cost of $N \times (9|d|^2 \times \text{ATTN} + 3|d| \times \text{FFN})$ for DyPRAG and $M \times (9|d|^2 \times \text{ATTN} + 3|d| \times \text{FFN})$ for PRAG, where $N$ represents the size of the training dataset $\mathcal{K}$ and $M$ denotes the size of the test set. The common divisor of offline parametrization is $E_1 \times (81|d|^2 \times \text{ATTN} + 9|d| \times \text{FFN})$, where $E_1$ is the number of epochs for LoRA training.

Additionally, to train our $\mathcal{F}'_\phi$ for $E_2$ epochs, we need to perform both forward and backward passes (the backward pass requires twice the cost of the forward pass) on one QA pair and its corresponding document in each step. This results in a cost of $N \times E_2 \times 9(|qa|+|d|)^2 \times \text{ATTN} + 3(|qa|+|d|) \times \text{FFN}$, with a negligible cost for translation. As shown in Figure 6 and 7, our DyPRAG achieves stable results with as few as 480 examples (even fewer is powerful), while $M = 3000$ in our experiments, and this value would be significantly larger in real-world applications.

For instance, using LLaMA3-8B as the backbone, producing a $\mathbf{P}_i$ requires 88 seconds, while one step for $\mathcal{F}'_\phi$ only takes an average of 15 seconds. Therefore, the total cost for training (excluding augmentation) is $M \times 88s$ in PRAG and $N \times 103s$ in DyPRAG. Assuming $N = 480$ and $M = 3000$, DyPRAG is 5.34x faster than PRAG. The low requirement for a large $N$ makes DyPRAG highly effective and generalizable for real-world scenarios, with extremely low costs that can be handled during offline training.

**Storage Cost.** As illustrated in Section A, each $\mathbf{P}_i$ requires 3.36MB for PRAG using Qwen2.5-1.5B, resulting in a total storage cost of 9.33GB in our main experiment. However, we significantly reduce this cost by imitating the underlying function between the document and parameters. Notably, the cost for $\mathbf{P}_i$ is a temporary cost in DyPRAG, which can be removed after collecting data or training one $\mathbf{P}_i$ and then updating $\mathcal{F}'_\phi$ by one step. Consequently, the overall cost of DyPRAG is substantially lower than that of PRAG (e.g., DyPRAG achieve better performance with only 7.71MB of storage as shown in Table 11).

# B EXPERIMENT SETUP

## B.1 IMPLEMENTATION DETAILS

**QA Datasets.** To ensure a comprehensive evaluation, we assess our method using the following datasets:

- **2WikiMultihopQA (2WQA)** (Ho et al., 2020) is designed to evaluate a model's capability in multi-hop reasoning by synthesizing information from multiple Wikipedia passages.
- **HotpotQA (HQA)** (Yang et al., 2018) similarly targets multi-hop reasoning, requiring models to amalgamate information from various contexts to answer a single query.
- **PopQA (PQA)** (Mallen et al., 2022) focuses on factual question answering, posing challenges that test the model's ability to recall precise knowledge and navigate ambiguities in entity representation.
- **ComplexWebQuestions (CWQ)** (Talmor & Berant, 2018) entails answering complex, multi-step questions sourced from the web, further challenging the model's capacity to retrieve and reason over extensive web content.

**Offline Doc-Param Pairs Collection.** Following (Jiang et al., 2023; Su et al., 2025), we utilize Wikipedia dumps as the external knowledge corpus, adopting the dataset proposed by DPR (Karpukhin et al., 2020). For document augmentation, each document is rewritten once, and three QA pairs are generated based on the document. Unless explicitly stated otherwise, the downstream LLM is used

for this purpose. During LoRA fine-tuning, the learning rate was set to $3 \times 10^{-4}$, and training was conducted for a single epoch (except PQA for 2). The LoRA modules were integrated exclusively into the feed-forward network (FFN) matrices, while the query, key, and value (QKV) matrices were excluded. The scaling factor $\alpha$ was set to 32, the LoRA rank $r$ was configured to 2, and no dropout was applied to ensure training stability and maximize parameter updates. The LoRA weights were randomly initialized following the settings outlined in the original LoRA paper (Hu et al., 2022).

**Baselines Implementation.** To conduct comprehensive experiments, we compare our DyPRAG with two commonly used baselines: SFT and Context-DPO, alongside parametric and non-parametric RAG baselines. For SFT, widely regarded as a standard approach for adapting models to various downstream tasks, is included to evaluate the generalization ability of DyPRAG. Specifically, we use the exact same hyperparameters as DyPRAG, setting the learning rate to $3 \times 10^{-4}$, fine-tuning on the same dataset (i.e., 36,000 samples) with a batch size of 1 for 1 epoch. For Context-DPO, we follow the implementation described in Bi et al. (2024). To ensure a fair comparison, we configure the trainable LoRA modules for both methods to match those in DyPRAG, maintaining equivalent parameter learning capacity. The LoRA modules are integrated exclusively into the FFN, while the query, key, and value matrices are excluded. The scaling factor $\alpha$ is set to 32, and the LoRA rank $r$ is configured as 2.

**Inference Settings.** All experiments use the publicly available Hugging Face implementations of LLaMA and Qwen. To ensure fairness, DyPRAG and all baselines share the same prompt template in Figure 11 and 12 following Su et al. (2025) and adopt of greedy decoding for result reproducibility. The max number of new tokens is set to 128.

**Retrieval Module $\mathcal{R}$.** Recent research on retrieval-augmented generation (RAG) (Ram et al., 2023) has shown that BM25 matches or even surpasses state-of-the-art dense retrieval models in certain scenarios. Following Su et al. (2025), we adopt BM25 as the retriever in our approach and Elasticsearch is used as the backend for implementing BM25.

**Training $\mathcal{F}'_\phi$.** Motivated by Liao et al. (2024a), we use simple MLP hypernetwork to transform embedding into adapter parameters. Through cross validation, the learning rate was set to $1 \times 10^{-5}$, and the training epoch was set to 1 which making the overall alignment process quickly. The truncation max length of text is set to 3000, which is larger than most retrieved documents. The performance reports for Qwen2.5-1.5B and LLaMA3.2-1B in Table 1 are based on training with 4,800 examples, while LLaMA3-8B is trained on 2,400 examples (except for 480 examples on 2WQA).

**Implementation of OOD Experiment.** To evaluate the generalization ability of our proposed DyPRAG, we select to out-of-distribution (OOD) datasets to conduct.

- **StrategyQA (SQA)** (Geva et al., 2021): A QA benchmark where reasoning steps are implicit in the question and must be inferred through strategic reasoning, including human-curated evidence paragraphs from Wikipedia.
- **IIRC** (Ferguson et al., 2020): A dataset comprising over 13,000 questions based on English Wikipedia paragraphs that provide only partial information and supplemented with samples from SQuAD 2.0 (Rajpurkar et al., 2016) and DROP (Dua et al., 2019), requiring retrieval of missing details from linked documents.
- **OpenBookQA (OBQA)** (Mihaylov et al., 2018): A multiple-choice QA dataset derived from a subset of WorldTree (Jansen et al., 2018), mainly focus on common knowledge.
- **MedMCQA (MQA)** (Pal et al., 2022): A multiple-choice QA dataset designed to address real-world medical domain entrance exam questions.
- **CNNDailymail**[1]: A summarization datasets containing just over 300k unique news articles as written by journalists at CNN and the Daily Mail.

For each dataset, we select the first 300 examples for testing and evaluate performance using F1 score for IIRC, Accuracy for SQA, Recall for OBQA and MQA and Rouge-L (Lin, 2004) for

---

[1] https://huggingface.co/datasets/ccdv/cnn_dailymail

Table 8: The experimental results of DyPRAG are compared with other effective RAG methods. All metrics are reported as F1 scores (%). The best performance is bolded, while the second-best is underlined. The evaluation is conducted on 2WQA and HQA datasets, focusing exclusively on the total sub-task.

| Base LLM | Method | 2WQA Total | HQA Total | Avg |
|---|---|---|---|---|
| | **RAG** | 23.12 | **27.14** | 25.13 |
| | **DRAGIN** | 21.73 | 12.50 | 17.12 |
| LLaMA3.2-1B | **FLARE** | 21.55 | 19.38 | 20.47 |
| | **DyPRAG (ours)** | 25.31 | 19.97 | 22.64 |
| | **DyPRAG-Combine (ours)** | **29.18** | 26.58 | 27.88 |
| | **RAG** | 24.31 | 20.73 | 22.52 |
| | **DRAGIN** | 25.01 | 8.51 | 16.76 |
| Qwen2.5-1.5B | **FLARE** | 21.56 | 7.97 | 14.77 |
| | **DyPRAG (ours)** | **26.46** | 19.67 | 23.07 |
| | **DyPRAG-Combine (ours)** | 25.18 | **27.57** | 26.38 |
| | **RAG** | 34.55 | 24.23 | 29.39 |
| | **DRAGIN** | 35.69 | 12.16 | 23.93 |
| LLaMA3-8B | **FLARE** | 34.62 | 29.43 | 32.03 |
| | **DyPRAG (ours)** | 37.25 | 22.55 | 29.90 |
| | **DyPRAG-Combine (ours)** | **45.17** | **38.35** | **41.76** |

Table 9: The experimental results of DyPRAG are compared with standard RAG based on Qwen3-8B and Qwen3-4b-Instruct. All metrics are reported as EM scores (%) and F1 scores (%). The best performance is bolded, while the second-best is underlined. The **Avg** is the average performance over all tasks.

| Base LLM | Method | 2WQA (Total) | | HQA (Total) | | PQA | | CWQ | | Avg | |
|---|---|---|---|---|---|---|---|---|---|---|---|
| | | EM | F1 | EM | F1 | EM | F1 | EM | F1 | EM | F1 |
| | **Vanilla** | 24.67 | 31.33 | 21.00 | 28.12 | 0.00 | 0.40 | 22.33 | **36.01** | 17.00 | 23.97 |
| Qwen3-8B | **RAG** | **35.33** | **42.26** | **32.33** | **44.00** | 0.33 | 9.17 | **23.00** | 35.79 | **22.75** | **32.81** |
| | **DyPRAG** | 21.00 | 27.94 | 20.33 | 27.82 | 0.00 | 0.46 | 17.67 | 29.12 | 14.75 | 21.34 |
| | **DyPRAG-Combine** | 31.00 | 38.37 | 29.67 | 39.81 | **0.33** | 4.54 | 20.00 | 31.37 | 20.25 | 28.52 |
| | **Vanilla** | 21.00 | 28.97 | 15.00 | 23.32 | 8.67 | 12.10 | 0.00 | 1.58 | 11.17 | 16.49 |
| Qwen3-4B-Instruct | **RAG** | 25.67 | 32.81 | 25.33 | 36.62 | 18.67 | 26.32 | 2.00 | 7.36 | 17.92 | 25.78 |
| | **DyPRAG** | 27.00 | 35.44 | 16.33 | 24.00 | 10.00 | 13.49 | 0.33 | 4.47 | 13.42 | 19.35 |
| | **DyPRAG-Combine** | **31.00** | **38.37** | **29.67** | **39.81** | **20.67** | **27.33** | **8.67** | **19.14** | **22.50** | **31.16** |

CNNDailymail (except for 100 examples) as metrics. Both datasets provide with ground-truth passages which indicate a more rigorous evaluation setting. For IIRC, we adopt the few-shot prompts from Su et al. (2024), while SQA, OBQA, MQA and CNNDailymail are evaluated in a zero-shot setting. Notably, the same prompt format (in Figure 11 and 12) from the main experiment is used to ensure a fair comparison, expect CNNDailymail using summarization template in Figure 13.

**Implementation of RAGTruth Experiment.** RAGTruth (Niu et al., 2023) is a benchmark dataset designed to evaluate the extent of hallucination in models. For our evaluation, we randomly select 100 QA-type subsets from RAGTruth, ensuring alignment with the training data of $\mathcal{F}'_\phi$. Notably, some questions in RAGTruth require the provided documents to be answerable which are more difficult. Interestingly, during evaluation, we observe that $\mathcal{F}'_\phi$ with fewer trained parameters perform better in such scenarios. Specifically, we train only 480 examples for LLaMA3.2-1B and Qwen2.5-1.5B, and 240 examples for LLaMA3-8B. We use GPT-4o as judge using prompt template in Figure 14.

# C SUPPLEMENT EXPERIMENT RESULTS

**Comparison with effective RAG baselines.** To compare our DyPRAG with effective RAG methods, we introduce two powerful baselines:

- FLARE (Jiang et al., 2023) is a multi-round retrieval augmentation method that triggers retrieval whenever it encounters an uncertain token. The query is defined as the last generated sentence excluding the uncertain tokens.

- DRAGIN (Su et al., 2024) improves multi-round retrieval by triggering only when an uncertain token has semantic significance and strongly influences subsequent tokens. It formulates queries using the model's internal state and preceding context.

The experimental results are presented in Table 8. Compared to standard RAG, DRAGIN and FLARE do not demonstrate significant performance advantages when the model size is smaller (e.g., LLaMA3.2-1B and Qwen2.5-1.5B). However, as the model size increases (e.g., LLaMA3-8B), DRAGIN achieves the best performance on the 2WQA dataset, while FLARE performs best on the HQA dataset comparing with RAG baseline. This indicates that effective RAG methods are often constrained by the model's inherent capabilities and lack robust generalization. In contrast, our proposed DyPRAG consistently delivers superior performance in most cases, demonstrating the effectiveness of our approach. Furthermore, when combined with in-context injection, DyPRAG achieves an average improvement of 6.54% over standard RAG, highlighting the substantial potential of integrating parametric knowledge with contextual knowledge.

**OOD Performance in Summarization Task.** Since our training primarily focused on QA data, we are curious whether the parameter translator $\mathcal{F}'_\phi$ can generalize effectively to other tasks. To evaluate DyPRAG's performance on non-QA tasks, we conducted additional experiments using the CNNDailymail dataset for a summarization task. This task employed the prompt template outlined in Figure 13 and was evaluated using the Rouge-L (Lin, 2004). As presented in Table 10, we utilized the parameter translator to first transform documents into parametric knowledge, enhancing the overlap between knowledge and model inputs. This step encouraged LLMs to better leverage document-specific knowledge. Across all model scales, our DyPRAG-Combine approach achieved an average improvement of 0.21 in performance, demonstrating that our method is effective beyond QA tasks. This result highlights the capability of our approach to perform general mapping from textual embeddings to the parametric space.

**Comparison of Response Length.** Notably, we consider only the context length when calculating inference cost. However, in practice, the response length from LLMs also affects inference time. As shown in Figure 4, we compare DyPRAG-Combine with RAG across four benchmarks, considering the average response length. DyPRAG-Combine significantly reduces response length, by 20% in 2WQA and up to 90% in CWQ. This demonstrates that DyPRAG-Combine can answer questions correctly with fewer tokens, thereby lowering inference costs and avoiding redundant information.

**Performance of DyPRAG on Non-Instruct Models.** With the rapid advancement of reinforcement learning, a growing number of long-context models, referred to as large reasoning models (LRMs) have emerged (Guo et al., 2025; Yang et al., 2025). Our goal is to evaluate whether the current design of DyPRAG can adapt effectively to such up-to-date models. For this purpose, we selected Qwen3-8B[2] (a reasoning model) and Qwen3-4B-Instruct[3] (an instruct model) for experiments. As shown in Table 9, the performance of Qwen3-8B decreases significantly when DyPRAG generated parameters are applied. This decline is primarily due to differences in answer patterns. LRMs tend to generate extremely lengthy reasoning trajectories, whereas our method only augments simple and short QA pairs. In contrast, the results for the instruct model, Qwen3-4B-Instruct, align with our main experiments, demonstrating that the current method is well-suited for instruct models. To enable compatibility with LRMs, the parameter translation process needs to be integrated into the reinforcement learning training pipeline. Addressing this challenge will be a focus of our future work.

# D  ADDITIONAL ABLATION EXPERIMENT RESULTS

**Effect of Training Dataset Size.** We adjust the pre-selected size of the training dataset composed of Doc-Param pairs, increasing it from 480 to 4800. As shown in Figure 6 and 7, DyPRAG achieves strong performance even with just 480 training examples. The performance remains remarkably stable

---

[2]https://huggingface.co/Qwen/Qwen3-8B

[3]https://huggingface.co/Qwen/Qwen3-4B-Instruct-2507

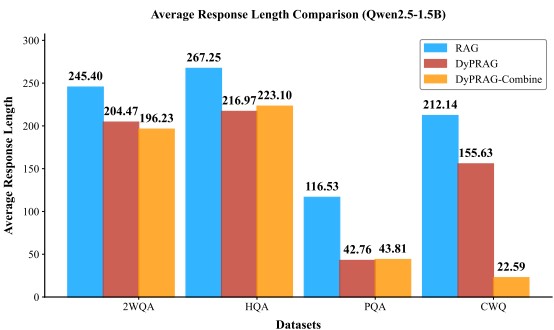

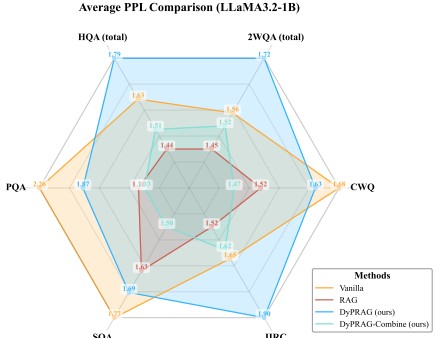

Figure 4: Comparison of response length across various datasets. The backbone model is the Qwen2.5-1.5B.

Figure 5: Comparison of average PPL. Smaller PPL means less conflicts. The backbone model is the LLaMA3.2-1B.

Table 10: The OOD performance on summarization dataset CNNDailymail. The metric is reported as Rouge-L (%) (Lin, 2004).

| Base LLM | Method | CNNDailymail |
|---|---|---|
| LLaMA3.2-1B | RAG | 21.09 |
| | DyPRAG-Combine | **21.21** |
| Qwen2.5-1.5B | RAG | 19.86 |
| | DyPRAG-Combine | **20.38** |
| LLaMA3-8B | RAG | **22.34** |
| | DyPRAG-Combine | 22.33 |

Table 11: Ablation study of intermediate dimension $p$ of $\mathcal{F}'_\phi$. The backbone model is the Qwen2.5-1.5B. The inference time is computed by average time of CWQ with batch_size of 1. The encode time is highlighted in red, while the translate time is marked in blue.

| Method | CWQ F1 | Inference Time (s) | Storage Cost (MB) |
|---|---|---|---|
| Vanilla | 26.47 | 0.56 (0.47x) | - |
| RAG | 28.32 | 1.20 (1x) | - |
| PRAG | 30.82 | 0.56 (0.47x) | 19107.84 (1x) |
| DyPRAG ($p = 2$) | 32.66 | 0.56+0.13+0.060 (0.625x) | 7.71 (0.04%x) |
| DyPRAG ($p = 4$) | 33.26 | 0.56+0.13+0.062 (0.627x) | 15.42 (0.08%x) |
| DyPRAG ($p = 16$) | 32.08 | 0.56+0.13+0.055 (0.621x) | 61.70 (0.32%x) |
| DyPRAG ($p = 32$) | 31.94 | 0.56+0.13+0.060 (0.625x) | 123.39 (0.64%x) |

across different dataset sizes, indicating that our design, $\mathcal{F}'_\phi$, is capable of learning the underlying mapping between documents and parameters with minimal data.

**Performance Effect of Retrieved Documents Number.** For standard RAG, the number of retrieved documents, denoted as $c$, is a crucial hyperparameter to tune. Recent studies (Leng et al., 2024; Wei et al., 2024) have investigated the impact of longer context lengths on standard RAG. As shown in Figure 8, the performance fluctuates as the number of retrieved documents increases, with the best value generally achieved at $c = 3$. This demonstrates that introducing more less-relevant context can negatively impact the model's ability to extract key information.

However, the effect of the number of injected documents in parametric form remains underexplored. Our proposed DyPRAG framework can seamlessly adapt to this scenario due to its inherent flexibility. As shown in Figure 9, the performance of DyPRAG does not significantly improve as the number of injected documents increases. For instance, in the 2WQA and CWQ datasets, the best performance is achieved when using only the top-1 document. This indicates that the most relevant document, as determined by the retriever $\mathcal{R}$, is sufficient to provide the knowledge needed to answer the question effectively. On the other hand, in datasets such as HQA and PQA, the best performance is observed when $c = 3$, suggesting that when more relevant information is retrieved, simple averaging of LoRA parameters can effectively integrate the knowledge. Additionally, in three out of four datasets (except PQA), the model's performance declines when too many documents are injected. This observation aligns with the findings in Shi et al. (2023), which suggest that task-irrelevant redundant information can degrade the model's performance, especially the compression of documents is lossy.

**Computation Effect of Injected Documents Number.** We have specifically designed the code of DyPRAG to enable the rapid loading of document-specific LoRA modules, ensuring minimal delays during operation. As demonstrated in Table 12, DyPRAG achieves superior inference efficiency compared to standard RAG, particularly as the number of injected documents increases. While the inference time of standard RAG grows significantly with more injected documents, DyPRAG-

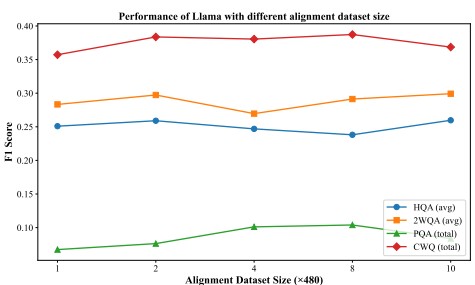

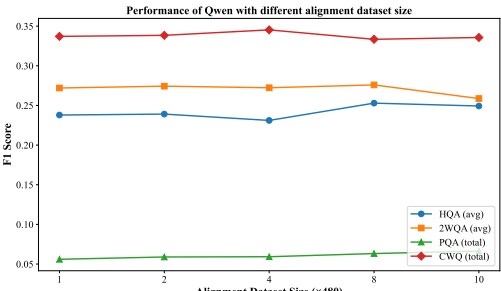

Figure 6: Ablation study of varying training dataset size for DyPRAG. The backbone model is the LLaMA3.2-1B.

Figure 7: Ablation study of varying training dataset size for DyPRAG. The backbone model is the Qwen2.5-1.5B.

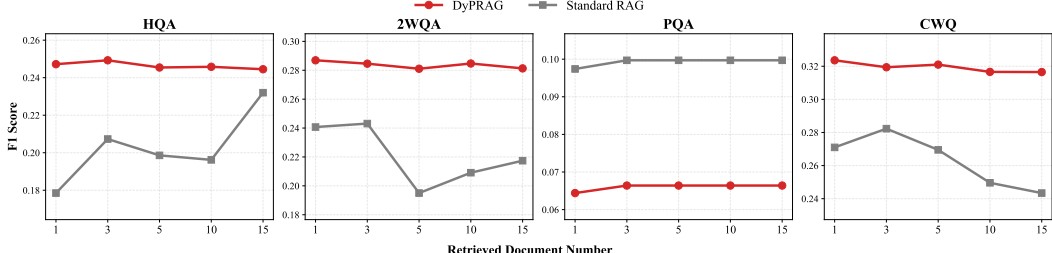

Figure 8: Ablation study of varying number of retrieved documents to RAG's performance. The backbone model is the Qwen2.5-1.5B.

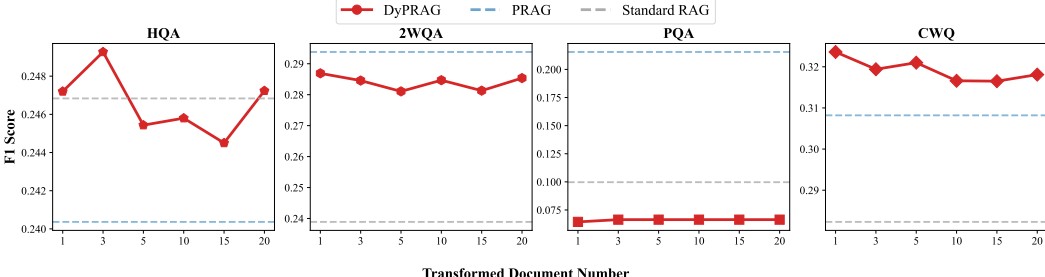

Figure 9: Ablation study of varying number of injected documents to DyPRAG's performance. The backbone model is the Qwen2.5-1.5B.

Combine maintains consistently lower inference times due to its shorter response lengths (as shown in Figure 4). Furthermore, DyPRAG significantly outperforms standard RAG in inference time which passages are excluded from the context.

However, the design of DyPRAG introduces an increase in encoding and translation time as the injected documents number grows. Currently, the encoding and translation processes are not fully optimized. In real-world applications, RAG-based queries are typically managed via message queues (e.g., Kafka (Kreps et al., 2011)), which provide a natural asynchronous execution environment. By leveraging this architecture, document embeddings can be extracted and transformed during the waiting period using separate process and model instance. Consequently, when the query reaches the processing stage, there is no additional encoding or translation delay. This allows DyPRAG to outperform standard RAG in both performance and inference efficiency.

Table 12: Ablation study of varying number of injected documents to computation cost. The backbone model is the Qwen2.5-1.5B.

| Documents | Method | Inference Time | Loading Time | Translate Time | Encode Time |
|---|---|---|---|---|---|
| 3 | DyPRAG | **0.84** | 0.0037 | 0.056 | 0.132 |
| | RAG | 1.23 | - | - | - |
| | DyPRAG-Combine | **0.36** | 0.0037 | 0.055 | 0.132 |
| 10 | DyPRAG | **0.80** | 0.0044 | 0.185 | 0.433 |
| | RAG | 1.54 | - | - | - |
| | DyPRAG-Combine | **0.78** | 0.0045 | 0.185 | 0.432 |
| 20 | DyPRAG | **0.80** | 0.0057 | 0.361 | 0.862 |
| | RAG | 1.74 | - | - | - |
| | DyPRAG-Combine | **1.40** | 0.0057 | 0.361 | 0.862 |
| 30 | DyPRAG | **0.80** | 0.0067 | 0.545 | 1.295 |
| | RAG | 2.18 | - | - | - |
| | DyPRAG-Combine | **1.96** | 0.0067 | 0.545 | 1.294 |

Table 13: Ablation study of retriever. All metrics are reported as EM scores (%) and F1 scores (%). The backbone model is the LLaMA3-8B.

| Method | Retriever | 2WQA | | HQA | | PQA | | CWQ | | Avg | |
|---|---|---|---|---|---|---|---|---|---|---|---|
| | | EM | F1 | EM | F1 | EM | F1 | EM | F1 | EM | F1 |
| Vanilla | None | 30.00 | 36.43 | 19.89 | 28.64 | 4.67 | 7.96 | 30.00 | 42.44 | 24.43 | 31.85 |
| RAG | Sparse | 28.40 | 34.20 | 19.13 | 28.67 | 5.67 | 16.13 | 25.33 | 35.45 | 23.04 | 30.86 |
| | Dense | 22.20 | 27.82 | 11.66 | 20.60 | 6.33 | 12.15 | 24.67 | 36.48 | 17.70 | 24.95 |
| DyPRAG | Sparse | 32.07 | 39.17 | 24.67 | 37.33 | 11.00 | 13.60 | 32.67 | 41.87 | 27.80 | 36.23 |
| | Dense | 22.20 | 28.48 | 15.67 | 23.34 | 8.33 | 11.09 | 30.33 | 41.10 | 19.67 | 26.46 |
| DyPRAG-Combine | Sparse | 36.33 | 47.68 | 33.22 | 43.22 | 21.00 | 32.86 | 29.67 | 39.07 | 33.20 | 43.69 |
| | Dense | 23.73 | 28.92 | 11.44 | 21.10 | 7.33 | 14.85 | 26.67 | 38.97 | 18.70 | 26.17 |

**Effect of Different Retriever.** Retrieval plays a critical role in RAG by determining whether the retrieved documents contain the necessary information to answer a given question. In the field of information retrieval, the two dominant retrieval methods are lexical matching(Robertson et al., 2009) and dense retrieval(Su et al., 2024). Among lexical matching techniques, BM25 stands out for its widespread adoption and proven effectiveness. In contrast, despite advancements in dense retrieval methods, none have achieved the same level of popularity or reliability as BM25. To explore the performance of these approaches, we employ the well-known all-MiniLM-L6-v2[4] model as the dense retriever which maps sentences into a 384-dimensional dense vector space, enabling dense retrieval tasks. As shown in Table 13, our experiments reveal that BM25 consistently outperforms dense retrieval methods across various datasets within the DyRPAG framework, despite the dense retrieval methods often excel in many other information retrieval tasks. These findings align with prior research (Su et al., 2024; Ram et al., 2023), which highlights BM25's robustness and effectiveness in RAG tasks. Despite significant advancements in dense retrieval technologies, our results reaffirm that the simpler, lexicon-based BM25 algorithm remains a strong baseline for improving LLM performance in RAG tasks.

# E EXPLORING METRICS FOR KNOWLEDGE CONFLICTS DETECTION

**Can Perplexity Reflects Knowledge Conflicts?** Recent studies have explored methods to detect hallucinations in LLMs and RAG systems by leveraging various metrics (Chen et al., 2024; Sun et al., 2024). Among these, we first adopt the simplest yet effective metric which only need single generation, Perplexity (PPL) (Ren et al., 2022), to evaluate knowledge conflicts. As illustrated in Figure5, Vanilla and DyPRAG exhibit higher PPL, while DyPRAG-Combine and RAG demonstrate significantly lower PPL. However, these results are inconsistent with the findings in Table 1 and Table 2. For instance, although DyPRAG-Combine achieves the best performance on IIRC, its calculated PPL suggests a higher probability of knowledge conflicts, which is clearly incorrect. We hypothesize that this discrepancy primarily stems from variations in model parameters introduced by parameter injection in DyPRAG, which cannot be detected using the simple PPL method. Given

---

[4]https://huggingface.co/sentence-transformers/all-MiniLM-L6-v2

Table 14: Textual similarity matrix (%) across both IID and OOD datasets. This matrix is computed based on the hidden states of retrieved documents and is symmetrical. It exhibits a significantly different trend when comparing documents from two distinct sources.

| Dataset | 2WQA | HQA | PQA | CWQ | IIRC | SQA | RAGTruth | OBQA | CNNDailymail | MQA |
|---|---|---|---|---|---|---|---|---|---|---|
| **2WQA** | 100 | 99.6 | 99.7 | 99.2 | 95.1 | 85.2 | 26.9 | 83.1 | 87.9 | 84.9 |
| **HQA** | – | 100 | 99.6 | 99.6 | 95.2 | 85.6 | 27.1 | 83.7 | 88.1 | 85.4 |
| **PQA** | – | – | 100 | 99.3 | 95.9 | 86.1 | 28.2 | 83.6 | 88.7 | 86.1 |
| **CWQ** | – | – | – | 100 | 94.5 | 84.3 | 26.3 | 84.1 | 86.6 | 84.8 |
| **IIRC** | – | – | – | – | 100 | 90.7 | 44.5 | 88.9 | 91.9 | 92.6 |
| **SQA** | – | – | – | – | – | 100 | 38.6 | 72.2 | 95.9 | 88.1 |
| **RAGTruth** | – | – | – | – | – | – | 100 | 51.3 | 40.4 | 54.3 |
| **OBQA** | – | – | – | – | – | – | – | 100 | 74.6 | 90.0 |
| **CNNDailymail** | – | – | – | – | – | – | – | – | 100 | 88.8 |
| **MQA** | – | – | – | – | – | – | – | – | – | 100 |

that different tokens contribute unequally to the overall semantics of a sentence, the PPL, which is calculated as the average of token-level uncertainty, fails to effectively capture the uncertainty of the entire sequence.

**Effective Detection with Sentence-Level Metrics.** Given the limitations of PPL, we decided to explore alternative metrics that leverage multiple generations. Research has shown that generating multiple outputs for a single input is beneficial for estimating sequence-level uncertainty. To this end, we set the temperature to 1.0, top_p to 0.95, and top_k to 20, generating five responses to calculate Entropy (EN), Length Normalized Entropy (LEN) (Malinin & Gales, 2020), and Lexical Similarity (LS) (Lin et al., 2022) to evaluate the probability of knowledge conflicts[5]. As shown in Table 5, our approach demonstrates reduced knowledge conflicts in most scenarios, especially in our strongest DyPRAG-Combine.

We observed that both EN and LEN increase when in-context injection is applied, suggesting that in RAG systems, the retrieved passages often conflict with the model's internal knowledge. In contrast, utilizing DyPRAG to inject converted parametric knowledge significantly reduces the likelihood of knowledge conflicts, demonstrating the effectiveness of DyPRAG. However, the LS results indicate that adding context reduces conflicts, which contradicts the established definition of knowledge conflicts. We argue that EN and LEN are more suitable for effective knowledge conflicts detection in DyPRAG settings. Exploring more effective detection methods remains an important direction for future work.

# F  DIVING INTO GENERALIZATION ABILITY OF DYPRAG

To train our parameter translator, we utilized datasets, including 2WikiMultihopQA, HotpotQA, PopQA, and ComplexWebQuestions. To evaluate generalization, we conducted OOD experiments on datasets such as IIRC, StrategyQA, RAGTruth, OpenBookQA, and CNNDailymail.

For IID datasets, the documents were retrieved exclusively from Wikipedia. In contrast, the OOD datasets exhibit diverse sources: IIRC primarily draws from English Wikipedia, supplemented with samples from SQuAD 2.0 (Rajpurkar et al., 2016) and DROP (Dua et al., 2019). StrategyQA includes human-curated evidence paragraphs from Wikipedia. RAGTruth is based on the QA set of MS MARCO (Nguyen et al., 2016), which originates from Bing search results. OpenBookQA is a multiple-choice QA dataset derived from a subset of WorldTree (Jansen et al., 2018). CNNDailymail is a summarization dataset comprising unique news articles authored by journalists at CNN and the Daily Mail.

To quantify the differences across datasets, we computed the vector similarity of the mean hidden states (i.e., the last-layer outputs of the final token) across them. As expected, the IID datasets exhibit extremely high similarity (>99%) due to their shared reliance on Wikipedia. In contrast, the OOD datasets show significantly lower similarity with the IID datasets. Although StrategyQA and IIRC

---

[5]We use the implementation in `https://github.com/alibaba/eigenscore`

Table 15: Parameter similarity matrix (%) across both IID and OOD datasets generated by parameter translator. This matrix is computed based on the generated parameters and is symmetrical. It exhibits a significantly different trend when comparing parameters from two distinct sources.

| Dataset | 2WQA | HQA | PQA | CWQ | IIRC | SQA | RAGTruth | OBQA | CNNDailymail | MQA |
|---|---|---|---|---|---|---|---|---|---|---|
| 2WQA | 89.15 | 88.27 | 85.29 | 88.19 | 88.49 | 87.66 | 47.15 | 83.16 | 89.25 | 86.39 |
| HQA | – | 83.45 | 84.84 | 85.50 | 84.66 | 84.95 | 49.55 | 82.99 | 86.27 | 84.29 |
| PQA | – | – | 88.55 | 86.88 | 88.57 | 86.31 | 47.45 | 82.40 | 87.98 | 85.78 |
| CWQ | – | – | – | 88.55 | 88.55 | 87.13 | 48.66 | 84.82 | 88.89 | 86.71 |
| IIRC | – | – | – | – | 90.43 | 87.58 | 47.93 | 82.88 | 88.48 | 86.34 |
| SQA | – | – | – | – | – | 89.01 | 48.87 | 84.75 | 89.63 | 86.75 |
| RAGTruth | – | – | – | – | – | – | 77.52 | 56.73 | 48.89 | 53.03 |
| OBQA | – | – | – | – | – | – | – | 92.95 | 86.02 | 88.56 |
| CNNDailymail | – | – | – | – | – | – | – | – | 91.97 | 88.47 |
| MQA | – | – | – | – | – | – | – | – | – | 89.42 |

primarily depend on Wikipedia, they include additional samples from other sources or incorporate human-curated content, which reduces their similarity to the IID datasets. Notably, RAGTruth demonstrates particularly low similarity, as its samples are carefully selected from MS MARCO to focus exclusively on content related to daily life. This underscores the substantial differences between the training corpora and our OOD evaluation datasets.

These findings further suggest that DyPRAG exhibits strong generalization capabilities, effectively adapting to the diverse characteristics of OOD datasets, as shown in Table 2.

## G    DOES PARAMETER TRANSLATOR REALLY LEARN TO GENERALIZE?

After obtaining the parameter translator, a natural question arises: does the parameter translator $\mathcal{F}'_\phi$ truly learn to generalize, or does it simply generate nearly identical LoRA matrices every time?

To investigate this, we collect 20 generated parameters across all datasets and compute the inter-average and intra-average parameter similarity. Since the parameter space itself is non-semantic, we measure similarity using the Frobenius norm: $1 - \frac{||A-B||_F}{\max(||A||_F, ||B||_F)}$. As shown in Table 15, the similarity of the generated LoRA parameters strongly correlates with the textual similarity of the inputs. In particular, the model produces significantly different outputs when exposed to distinct contexts, even from the same dataset. Although hypernetwork still lacks well-established interpretability methods, this simple comparison provides evidence that the hypernetwork is indeed mapping from different textual embeddings to diverse parameter space. We hope that future research will develop more comprehensive approaches to explain hypernetwork behavior.

## H    WHY VANILLA OUTPERFORMS RAG OCCASIONALLY?

In this section, we provide a detailed analysis of why the vanilla model occasionally outperforms RAG. As shown in Table 1, the vanilla model surpasses RAG most significantly in 2WQA, as the results vary across different models. For instance, the vanilla model outperforms RAG by 2.62% and 0.99% in Qwen2.5-1.5B and LLaMA3-8B on average in F1, respectively. After analyzing the cases, we identify two key issues that most affect RAG's performance: **1) Poor Retriever.** Following Su et al. (2025), we use BM25 as the retriever. However, in many cases, the retrieved documents contain only similar words rather than relevant content. This results in the provided content being unhelpful or even detrimental to LLMs. **2) Already Seen Data.** During the pre-training stages of the selected LLMs (Yang et al., 2024; Meta, 2024a;b), the external source we use (i.e., Wikipedia) has already been seen. This allows LLMs to answer certain questions independently, especially in simpler tasks like 2WQA. Moreover, the inclusion of incorrect or irrelevant context further degrades the performance, as observed in Table 1.

A more rigorous evaluation setting should include ground-truth passages and ensure no or less data leakage. Under this setting, as shown in Table 2, the performance of the vanilla model is significantly

lower than that of RAG, which aligns with our hypothesis. For instance, the vanilla model achieves only 8.78% and 1.00% accuracy on Qwen2.5-1.5B for IIRC and SQA, respectively. In contrast, DyPRAG demonstrates a notable improvement in test-time knowledge, achieving 10.23% and 15.67% accuracy on Qwen2.5-1.5B for IIRC and SQA, respectively. These results underscore the critical role of RAG while showcasing the ability of our proposed DyPRAG to seamlessly enhance OOD knowledge effectively. Furthermore, DyPRAG-Combine establishes a superior RAG paradigm by delivering even better performance under these more challenging conditions. In summary, we believe that this more rigorous experimental setting better validates our proposed method.

## I    FURTHER ANALYSIS OF CONTEXTUAL AND PARAMETRIC KNOWLEDGE CONFLICTS

**Parameter Injection Makes LLMs Trust Themselves.**    As shown in Table 16, while vanilla LLMs contain accurate parametric knowledge regarding which director was born later, the introduction of retrieved documents about each director causes contextual knowledge to mislead $\mathcal{M}$, resulting in the incorrect answer "William Lustig" while DyPRAG stays the same. This demonstrates that DyPRAG can effectively reduce the knowledge conflicts problem. In this case, standard RAG often introduces redundant or incorrect information from the context, a phenomenon commonly referred to as RAG hallucination (Sun et al., 2024). In contrast, our proposed DyPRAG effectively incorporates accurate information into parametric knowledge. This allows DyPRAG-Combine to align parametric knowledge with contextual knowledge, thereby reducing the likelihood of conflicts and enabling LLMs to rely more consistently on its own knowledge.

**Dynamic Parametric Knowledge Enhances LLMs at Test-time.**    Our DyPRAG serves as an effective plug-and-play technique for enhancing parametric knowledge during test-time. As demonstrated in Table 17, DyPRAG successfully manipulates the original parametric knowledge of LLMs in 14.67% of cases. Therefore, it can directly enhance the model's knowledge during inference without the need for further fine-tuning.

**Proportion of Different Combinations.**    Furthermore, as shown in Table 18, when both Vanilla LLMs and RAG give incorrect answers, DyPRAG provides the correct answer 26.33% of the time. This indicates that DyPRAG can effectively inject missing parametric knowledge and outperforms in-context injection methods. Additionally, in cases where the vanilla LLM provides the correct answer (i.e., the model possesses accurate internal knowledge), RAG achieves a correct answer rate of 5.33%, while DyPRAG performs better with a rate of 6.33%, showing that parameter injection leads to lower conflicts. Similar trend of DyPRAG-Combine is presented in Table 19.

These results demonstrate that our proposed DyPRAG injects parametric knowledge successfully and mitigates conflicts between internal parametric knowledge and external contextual knowledge through the injection of knowledgeable LoRA adapters.

Table 16: Case study about contextual and parametric knowledge conflicts in 2WQA (Bridge sub-task) where only standard RAG answers wrongly (6.67%). The backbone model is the LLaMA3.2-1B. ▉:deficiency in parametric knowledge, ▉: knowledge conflicts, ▉: successful knowledge manipulation.

| Question: Which film has the director born later, Diary Of A Maniac or Return Of The Hero ? | | |
|---|---|---|
| Ground truth: Return Of The Hero | | |
| Retrieved top-1 document: Maniac (1980 film) Maniac is a 1980 American psychological slasher film directed by William Lustig and written by C. A. Rosenberg... | | |
| **Method** | **Answer** | **Status** |
| **Vanilla** | Return Of The Hero | ✓ |
| **RAG** | William Lustig | ✗ |
| **DyPRAG** (ours) | Return Of The Hero | ✓ |
| **DyPRAG-Combine** (ours) | Return Of The Hero | ✓ |

Table 17: Case study about contextual and parametric knowledge conflicts in 2WQA (Bridge sub-task) where only DyPRAG and DyPRAG-Combine answer wrongly (14.67%). The backbone model is the LLaMA3.2-1B. ▉:deficiency in parametric knowledge, ▉: knowledge conflicts, ▉: successful knowledge manipulation

| Question: Which film has the director born later, Miss Sloane or Time Changer ? | | |
|---|---|---|
| Ground truth: Time Changer | | |
| Retrieved top-1 document: production budget of $13 million. " Miss Sloane " is ranked number 75 by per-theater average on Box Office... | | |
| **Method** | **Answer** | **Status** |
| **Vanilla** | John Frankenheimer | ✗ |
| **RAG** | Miss Sloane | ✗ |
| **DyPRAG** (ours) | Time Changer | ✓ |
| **DyPRAG-Combine** (ours) | Time Changer | ✓ |

Table 18: Right/Wrong answer combinations of Vanilla, RAG, DyPRAG and corresponding proportional distribution in 2WQA (Bridge Sub-task). The backbone model is the LLaMA3.2-1B. ✓ indicates a correct answer, while ✗ indicates an incorrect answer. The "Ratio (%)" column on the right represents the percentage of each combination across the dataset (300 examples).

| Vanilla | RAG | DyPRAG | Ratio(%) |
|---|---|---|---|
| ✓ | ✓ | ✓ | 4.67 |
| ✗ | ✗ | ✗ | 34.67 |
| ✓ | ✗ | ✓ | **6.33** |
| ✓ | ✓ | ✗ | 5.33 |
| ✗ | ✓ | ✓ | 8.33 |
| ✗ | ✗ | ✓ | **26.33** |
| ✗ | ✓ | ✗ | 7.67 |
| ✓ | ✗ | ✗ | 6.33 |

Table 19: Right/Wrong answer combinations of Vanilla, RAG, DyPRAG-Combine and corresponding proportional distribution in 2WQA (Bridge Sub-task). The backbone model is the LLaMA3.2-1B. ✓ indicates a correct answer, while ✗ indicates an incorrect answer. The "Ratio (%)" column on the right represents the percentage of each combination across the dataset (300 examples).

| Vanilla | RAG | DyPRAG-Combine | Ratio(%) |
|---|---|---|---|
| ✓ | ✓ | ✓ | 5.33 |
| ✗ | ✗ | ✗ | 35.00 |
| ✓ | ✗ | ✓ | **6.33** |
| ✓ | ✓ | ✗ | 4.67 |
| ✗ | ✓ | ✓ | 8.00 |
| ✗ | ✗ | ✓ | **26.00** |
| ✗ | ✓ | ✗ | 8.00 |
| ✓ | ✗ | ✗ | 6.67 |

Table 20: The experimental results of DyPRAG are compared with parametric RAG, standard RAG and two training-based methods. All metrics are reported as F1 scores (%). The best performance is bolded, while the second-best is underlined. The **Avg** is the average performance over all sub-tasks.

| Base LLM | Method | 2WQA | | | | | HQA | | | PQA | CWQ | Avg |
|---|---|---|---|---|---|---|---|---|---|---|---|---|
| | | Compare | Bridge | Inference | Compose | Total | Bridge | Compare | Total | | | |
| LLaMA3.2-1B | Vanilla | 42.89 | 24.17 | 16.91 | 7.87 | 22.52 | 13.25 | 40.26 | 18.79 | 2.26 | 34.94 | 22.39 |
| | SFT | 25.36 | 10.87 | 6.05 | 3.35 | 10.60 | 1.86 | 4.51 | 2.51 | 1.33 | 12.77 | 7.92 |
| | Context-DPO | 37.28 | 39.39 | 16.29 | 4.86 | 22.89 | 17.03 | 32.86 | 20.17 | 12.79 | 13.00 | 21.66 |
| | RAG | 41.23 | 26.78 | 22.51 | 10.21 | 23.12 | 21.38 | 42.46 | 27.14 | 17.65 | 37.39 | 26.99 |
| | PRAG | 50.20 | 24.34 | 19.11 | 8.24 | 27.73 | 13.65 | 40.90 | 21.50 | 23.58 | 35.86 | 26.51 |
| | PRAG-Combine | 40.50 | 31.30 | 22.85 | 9.77 | 30.30 | 22.56 | 41.55 | 28.31 | 32.59 | 39.63 | 29.94 |
| | DyPRAG (ours) | 51.25 | 48.15 | 17.35 | 7.54 | 25.31 | 14.05 | 43.90 | 19.97 | 11.33 | 36.86 | 27.57 |
| | DyPRAG-Combine (ours) | 52.13 | 46.19 | 22.54 | 12.60 | 29.18 | 22.05 | 43.78 | 26.58 | 29.93 | 38.96 | 31.80 |
| Qwen2.5-1.5B | Vanilla | 45.74 | 39.06 | 17.04 | 7.27 | 26.87 | 12.18 | 39.46 | 17.76 | 2.87 | 26.47 | 25.79 |
| | SFT | 37.98 | 43.44 | 9.06 | 3.83 | 18.75 | 5.82 | 26.21 | 8.85 | 6.95 | 13.96 | 17.49 |
| | Context-DPO | 35.01 | 40.59 | 17.88 | 6.51 | 21.78 | 19.12 | 31.41 | 22.51 | 14.18 | 19.20 | 22.82 |
| | RAG | 38.75 | 38.84 | 11.87 | 5.68 | 24.31 | 16.19 | 37.13 | 20.73 | 9.97 | 28.23 | 23.17 |
| | PRAG | 44.96 | 43.96 | 19.29 | 11.14 | 27.55 | 13.27 | 40.42 | 18.42 | 21.55 | 30.82 | 27.14 |
| | PRAG-Combine | 40.50 | 44.00 | 16.30 | 8.17 | 27.49 | 18.86 | 36.49 | 23.10 | 23.43 | 32.13 | 27.05 |
| | DyPRAG (ours) | 43.03 | 47.20 | 17.04 | 8.55 | 26.46 | 13.72 | 41.39 | 19.67 | 6.64 | 31.94 | 25.56 |
| | DyPRAG-Combine (ours) | 35.83 | 44.89 | 14.81 | 8.64 | 25.18 | 21.56 | 41.25 | 27.57 | 22.69 | 33.57 | 27.60 |
| LLaMA3-8B | Vanilla | 54.90 | 55.20 | 24.59 | 14.43 | 33.02 | 19.00 | 45.63 | 21.29 | 7.96 | 42.44 | 31.85 |
| | SFT | 9.66 | 26.26 | 16.79 | 1.12 | 11.63 | 2.04 | 2.48 | 2.05 | 0.00 | 5.92 | 7.80 |
| | Context-DPO | 46.90 | 25.57 | 20.83 | 6.81 | 21.98 | 13.86 | 32.97 | 18.17 | 18.68 | 13.81 | 21.96 |
| | RAG | 58.43 | 47.77 | 19.20 | 11.07 | 34.55 | 19.68 | 42.10 | 24.23 | 16.13 | 35.45 | 30.86 |
| | PRAG | 57.78 | 58.93 | 27.61 | 19.17 | 39.19 | 33.68 | 65.88 | 38.08 | 26.13 | 43.54 | 41.00 |
| | PRAG-Combine | 60.13 | 56.69 | 32.71 | 20.91 | 40.55 | 39.41 | 68.22 | 44.84 | 26.23 | 36.41 | 42.61 |
| | DyPRAG (ours) | 57.39 | 56.43 | 25.33 | 18.88 | 37.80 | 24.85 | 58.59 | 28.56 | 13.60 | 41.87 | 36.23 |
| | DyPRAG-Combine (ours) | 66.00 | 59.46 | 35.78 | 26.90 | 50.24 | 33.37 | 57.93 | 38.35 | 32.86 | 39.07 | 43.69 |

# J  VISUALIZATION OF PARAMETER TRANSLATOR WORKFLOW.

To clearly illustrate the workflow of the parameter translator $\mathcal{F}'_\phi$, we use the up-proj module in the FFN as an example, as shown in Figure 10. This visualization demonstrates the transformation of document embeddings into dynamic LoRAs, consistent with Eq. 4.

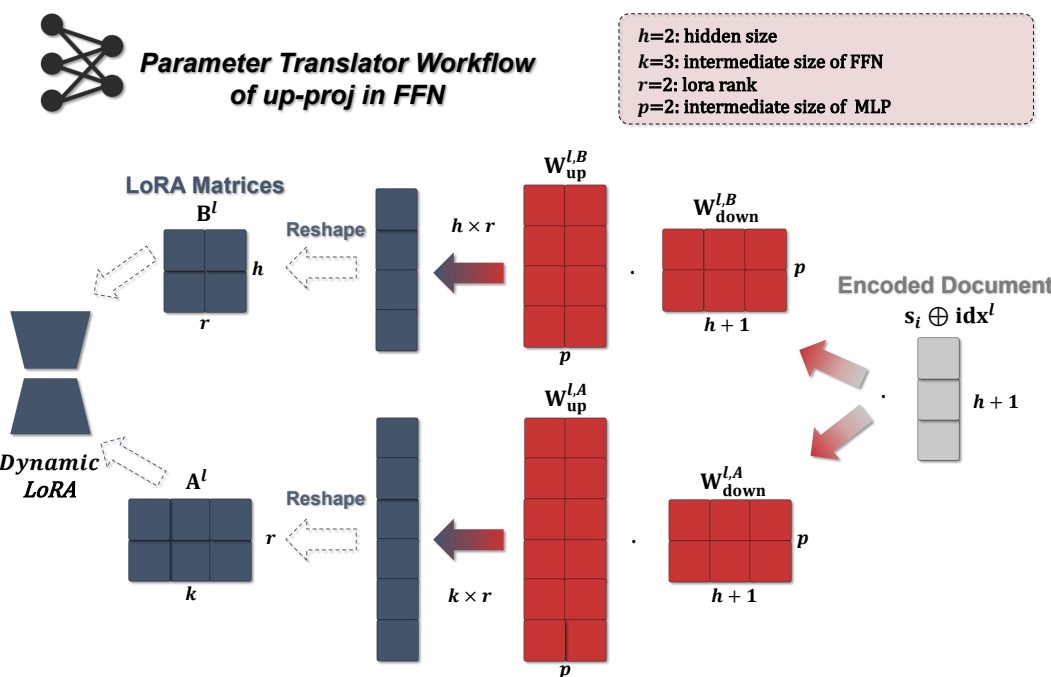

Figure 10: Visualization of the parameter translator workflow of up-proj in FFN. The overall process remains consistent with Eq. 4.

## K    PROMPT FOR MAIN EXPERIMENTS EVALUATION

In the main experiments, we used the following prompt to assess the performance of DyPRAG and other baseline models in Figure 11 and  12:

---

**Prompt Format of No-CoT**

You should answer the question by referring to the knowledge provided below and integrating your own knowledge.

Passage 1: {passages[0]}
Passage 2: {passages[1]}
Passage 3: {passages[2]}

Question: {question}
The answer is {answer}

---

Figure 11: Prompt format of No-CoT in our expriments.

---

**Prompt Format of CoT**

You should reference the knowledge provided below and combine it with your own knowledge to answer the question. Please follow the format of the example I provided above. Here are some examples about how to answer the questions.
Question: $\text{fewshot}_q[0]$
Answer: $\text{fewshot}_a[0]$
Question: $\text{fewshot}_q[1]$
Answer: $\text{fewshot}_a[1]$
Question: $\text{fewshot}_q[2]$
Answer: $\text{fewshot}_a[2]$
...

Here are some reference.
Passage 1: {passages[0]}
Passage 2: {passages[1]}
Passage 3: {passages[2]}

Let's think step by step. Answer the questions in the same format as above.
Question: {question}
Answer: {answer}

---

Figure 12: Prompt format of CoT in our expriments.

In summarization experiment in CNNDailymail, we used the following prompt to assess the performance of DyPRAG and other baseline models in Figure 13:

---

**Prompt Format of summarization**

Please summarize the main ideas from the content of Passage 1 in a clear and concise manner.

Passage 1: {passages[0]}

---

Figure 13: Prompt format of summarization in our expriments.

## L    PROMPT FOR KNOWLEDGE INTERNALIZATION EVALUATION

In the knowledge internalization experiments, we used the following prompt to assess the internalization ability of RAG generation from DyPRAG-Combine and RAG method evaluated by GPT-4o in Figure 14:

---

**Prompt Format of Evaluate RAGTruth**

Compare DyPRAG and RAG answers to assess which better internalizes knowledge—integrating its own knowledge with the given context for a natural, informed response.

Evaluation Criteria:
1. Internalization: Does the answer go beyond repetition to integrate knowledge seamlessly?
2. Fluency: Is the response well-structured and readable?
3. Relevance: Does it stay on topic while demonstrating depth?

Mark the Winner: Identify the superior response. If both are equally strong, mark it as a tie.

Question: {question}
Context: {passages}
DyPRAG Answer: {dyprag_answer}
RAG Answer: {rag_answer}

Respond in the following format:
{{
"win model": "DyPRAG or RAG or Tie",
"reason": "Provide a concise explanation of why the selected answer demonstrates better knowledge integration, referencing the question, context, and specific details from both answers. If one answer has clear advantages in integration, explain them; if there are errors or weaknesses, specify them."
}}

---

Figure 14: Prompt format of evaluate RAGTruth using GPT-4o. We compare answer between standard RAG and DyPRAG-Combine.

## M    FUTURE DIRECTIONS

In this study, our proposed Dynamic Parametric RAG (DyPRAG) demonstrates superior performance in both IID and OOD settings across various scales of LLMs. Developing and deploying such RAG system in real-world applications is a promising and worthwhile avenue for future work. Moreover, we believe the most promising direction for DyPRAG lies in the integration of memory (Wang et al., 2024d), which is commonly implemented using external textual databases. This raises a fundamental question: *Can a parameter translator convert any textual knowledge into parametric knowledge?* If the answer is yes, it would enable the replacement of large, text-based memory banks with a simple, plug-and-play memory translator. This approach opens up an exciting avenue for enhancing the fine-grained knowledge manipulation capabilities of $\mathcal{F}'_\phi$, which we aim to explore in future work.

## N    REPRODUCIBILITY

In this work, we use open-source LLMs and publicly available datasets to conduct our experiments. To ensure reproducibility, we provide the implementation details in Section 4.1 and Appendix B.1. Details of all prompts referenced in this paper are included in Appendix K and L. The full code and a detailed reproduction procedure of DyPRAG, which can be accessed via the following link: `https://anonymous.4open.science/r/DyPRAG_ICLR`. We also provide a well-trained parameter translator used in our experiments, available through the anonymous link.

## O    THE USAGE OF LLMS

We used Large Language Models (LLMs) to perform minor language polishing and grammar refinement on select sections of the paper. The LLMs were not involved in generating core content,

conducting research, or formulating ideas. All substantive contributions, including analysis, results, and conclusions, were independently produced by the authors.

