# OpenReview forum: "Dynamic Parametric Retrieval Augmented Generation"
_ICLR.cc/2026/Conference — ICLR 2026 Conference Withdrawn Submission_

### Official Review · Reviewer_ncDL · 2025-10-30

**Soundness:** 1
**Presentation:** 2
**Contribution:** 2
**Rating:** 2
**Confidence:** 4

**Summary:**

This paper presents Dynamic Parametric RAG (DyPRAG), a revision to Parametric RAG that generates parametric knowledge dynamically during test time. Given a set of retrieved documents relevant to the query, a trainable parameter translator will generate the corresponding parametric representation as dynamic LoRA for each retrieved document. Then, the summation of all the dynamic LoRA will be applied to the LLM to generate the final outcome. Experimental results on 2WikiMultihopQA, HotpotQA, PopQA, and ComplexWebQuestions show the effectiveness of the proposed method compared with PRAG, pure RAG and other baseline.

**Strengths:**

* An interesting revision to the PRAG framework.

* The current experiments show the improvement of performances from the proposed method.

**Weaknesses:**

* In experiments, only BM25 is used as the retrieval model, and the size of retrieval documents is fixed at 3. However, the performance of the baseline RAG could be dramatically different with a dense retrieval model and/or more reference documents. The pure sparse RAG is simplistic and efficient. A fair comparison could be made by fixing the computation cost. For example, the performances of pure RAG with more reference documents and even with dense retrieval could be reported. At the same level of computation cost, can the proposed method outperform the RAG or PRAG?

* In addition to the statement in Section 3.3 and Appendix A, empirical runtime in the inference stage could be measured and compared with other methods. In particular, the runtime of the best-performing setting, DyPRAG-Combine, should be reported and analyzed. Without the efficiency information, the practical usefulness of the proposed method is still unclear.

* This work claims that the traditional PRAG requires retraining when new documents added. As shown in Table 6, the performance of the proposed method in the OOD setting is very close to baseline RAG. RAG could possibly outperform the proposed method with a different settings (e.g., more reference documents, a different retrieval models).

* Figure 1, which is very important to this paper, could be better presented. The scale is too small in the current version.

**Questions:**

* Did you try different number of retrieved documents in the baseline RAG?

* How does the retrieval model affect the performance of the DyPRAG?

---

> ### Author Response · Authors · 2025-11-20
>
> Dear reviewer ncDL:
>
> Thank you for your sincere review and constructive feedback. We address your concerns as follows:
>
> > **W1.1: Performance with a dense retrieval model.**
>
> Thank you for raising this concern. Different retrievers can indeed affect the quality of retrieved documents. To evaluate the performance of dense retrieval, we adopt the well-known all-MiniLM-L6-v2 model as our dense retriever, which maps sentences into a 384-dimensional vector space. A complete analysis is provided in Appendix D, *Effect of Different Retrievers* (Table 13, also shown below).
>
> The results indicate that **BM25 consistently outperforms dense retriever** across all datasets under the DyPRAG framework. Across all settings, **DyPRAG also consistently surpasses RAG**, particularly in combination scenarios.
> Due to the extremely long indexing time required by dense retrievers, we defer experiments with stronger dense retrieval models (e.g., Qwen3) to later stages, where we will further evaluate their impact on both RAG and DyPRAG.
>
>
>
> | Method | Retriever | 2WQA | | HQA | | PQA | | CWQ | | Avg | |
> | :--- | :--- | :---: | :---: | :---: | :---: | :---: | :---: | :---: | :---: | :---: | :---: |
> | | | EM | F1 | EM | F1 | EM | F1 | EM | F1 | EM | F1 |
> | Vanilla | None | 30.00 | 36.43 | 19.89 | 28.64 | 4.67 | 7.96 | 30.00 | 42.44 | 24.43 | 31.85 |
> | RAG| Sparse | **28.40** | **34.20** | **19.13** | **28.67** | 5.67 | **16.13** | **25.33** | 35.45 | **23.04** | **30.86** |
> | | Dense | 22.20 | 27.82 | 11.66 | 20.60 | **6.33** | 12.15 | 24.67 | **36.48** | 17.70 | 24.95 |
> | DyPRAG| Sparse | **32.07** | **39.17** | **24.67** | **37.33** | **11.00** | **13.60** | **32.67** | **41.87** | **27.80** | **36.23** |
> | | Dense | 22.20 | 28.48 | 15.67 | 23.34 | 8.33 | 11.09 | 30.33 | 41.10 | 19.67 | 26.46 |
> | DyPRAG-Combine | Sparse | **36.33** | **47.68** | **33.22** | **43.22** | **21.00** | **32.86** | **29.67** | **39.07** | **33.20** | **43.69** |
> | | Dense | 23.73 | 28.92 | 11.44 | 21.10 | 7.33 | 14.85 | 26.67 | 38.97 | 18.70 | 26.17 |
>
> > **W1.2: Performance with more reference documents.**
>
> Thank you for your thoughtful comments. We analyzed this question for DyPRAG in Appendix D, *Performance Effect of Retrieved Document Number* (Figure 9). Following your suggestion, we also conducted similar experiments for RAG (Figure 8, also shown below). As illustrated, RAG’s performance fluctuates as the number of retrieved documents increases, with the best results generally obtained at **c = 3**, which is also our default setting. This indicates that **introducing additional less-relevant context can negatively affect the model’s ability to extract key information**.
>
> We observed a similar trend in DyPRAG; however, compared with RAG, DyPRAG is more robust to variations in document count and likewise achieves its optimal performance at **c = 3**.
>
> | Method | 2WQA (Total) | HQA (Total) | PQA | CWQ |
> | :--- | :---: | :---: | :---: | :---: |
> | RAG-1 | 24.07 | 17.85 | 9.74 | 27.10 |
> | RAG-3 | **24.31** | 20.73 | 9.97 | **28.23** |
> | RAG-5 | 19.5 | 19.86 | 9.97 | 26.95 |
> | RAG-10 | 20.91 | 19.62 | 9.97 | 24.96 |
> | RAG-15 | 21.74 | **23.2** | 9.97 | 24.34 |
>
> > **W2: Empirical runtime in the inference stage.**
>
> Thank you for raising this concern. In the previous version, we analyzed the end-to-end latency as the number of injected documents increased, and reported the results in the Appendix. Following your suggestion, we now include a short concluding remark at the end of Section 4.5, *Effect of Parameter Translator Size*, that points readers to the detailed analysis in Appendix D, *Computation Effect of Injected Documents Number* (Table 12).
>
> Our findings show that **DyPRAG achieves superior inference efficiency compared to standard RAG**, particularly as the number of injected documents increases. While the inference time of standard RAG grows substantially with more injected documents, DyPRAG-Combine consistently maintains lower inference latency, primarily due to its shorter response lengths (Figure 4). Moreover, when passages are excluded from the input context, DyPRAG still significantly outperforms standard RAG in inference time.
>
> However, the design of DyPRAG introduces additional encoding and translation overhead as the number of injected documents increases. At present, these encoding and translation procedures are not fully optimized. In real-world deployments, RAG-based queries are typically managed through message queueing systems (e.g., Kafka), which naturally support asynchronous execution. Under such an architecture, document embeddings can be computed and translated during the waiting period using separate processes or model instances, effectively mitigating this overhead.

---

> ### Author Response · Authors · 2025-11-20
>
> | Documents | Method          | Inference Time | Loading Time | Translate Time | Encode Time |
> |-----------|-----------------|----------------|--------------|----------------|-------------|
> | 3         | DyPRAG          | 0.84           | 0.0037       | 0.056          | 0.132       |
> |           | RAG             | 1.23           | -            | -              | -           |
> |           | DyPRAG-Combine  | 0.36           | 0.0037       | 0.055          | 0.132       |
> | 10        | DyPRAG          | 0.80           | 0.0044       | 0.185          | 0.433       |
> |           | RAG             | 1.54           | -            | -              | -           |
> |           | DyPRAG-Combine  | 0.78           | 0.0045       | 0.185          | 0.432       |
> | 20        | DyPRAG          | 0.80           | 0.0057       | 0.361          | 0.862       |
> |           | RAG             | 1.74           | -            | -              | -           |
> |           | DyPRAG-Combine  | 1.40           | 0.0057       | 0.361          | 0.862       |
> | 30        | DyPRAG          | 0.80           | 0.0067       | 0.545          | 1.295       |
> |           | RAG             | 2.18           | -            | -              | -           |
> |           | DyPRAG-Combine  | 1.96           | 0.0067       | 0.545          | 1.294       |
>
> > **W3: The performance in Table 6 is similar to RAG.**
>
> Thank you for this insightful question. The results in Table 6 (now Table 10) report theOOD performance on the CNNDailyMail summarization dataset, which is entirely different from our training setup. We include this dataset to examine whether our parameter translator can effectively inject document knowledge into the model **even for a completely unseen summarization task**, thereby helping the model internalize external information and produce better answers.
>
> On LLaMA3.2-1B and Qwen2.5-1.5B, we obtain improvements of **+0.12** and **+0.52**, respectively, and maintain comparable performance on LLaMA3-8B. These results demonstrate that our method provides clear benefits. As also validated in W1 and W2, our approach consistently delivers better performance under settings with more reference documents and different retrieval models. We attribute these gains to our ability to **proactively inject document knowledge into the model parameters, which reduces the discrepancy between contextual information and the model’s internal knowledge**.
>
> > **W4: The scale of  Figure 1 is too small.**
>
> Thank you for your sincere suggestion. We have updated our manuscript accordingly.
>
> > **Q1: See W1.2**
>
> > **Q2: See W1.1**
>
> **We have revised and improved our manuscript accordingly, adding more experiments in both the main text and the appendix. We sincerely welcome your further comments!**

---

> ### Author Response · Authors · 2025-11-26
>
> Dear Reviewer ncDL,
>
> To thoroughly evaluate **the impact of different retrievers as suggested in W1.1**, we have conducted additional experiments using the stronger **Qwen3-Embedding-0.6B model (1024-dimensional)** as our dense retriever.
>
> | Method | Retriever | 2WQA |  | HQA |  | PQA |  | CWQ |  | Avg |  |
> |--------|-----------|------|-------|-------|-------|------|-------|-------|-------|-------|-------|
> | | | EM | F1 | EM | F1 | EM | F1 | EM | F1 | EM | F1 |
> | Vanilla | None | 30 | 36.43 | 19.89 | 28.64 | 4.67 | 7.96 | 30 | 42.44 | 24.43 | 31.85 |
> | RAG | BM25 | 28.4 | 34.2 | 19.13 | 28.67 | 5.67 | 16.13 | 25.33 | 35.45 | 23.04 | 30.86 |
> | | all-MiniLM | 22.2 | 27.82 | 11.66 | 20.6 | 6.33 | 12.15 | 24.67 | 36.48 | 17.7 | 24.95 |
> | | Qwen | 19.53 | 25.86 | 12.11 | 19.57 | 6.63 | 13.79 | 25.67 | 36.36 | 16.63 | 23.82 |
> | DyPRAG | BM25 | 32.07 | 39.17 | 24.67 | 37.33 | 11 | 13.6 | 32.67 | 41.87 | 27.8 | 36.23 |
> | | all-MiniLM | 22.2 | 28.48 | 15.67 | 23.34 | 8.33 | 11.09 | 30.33 | 41.1 | 19.67 | 26.46 |
> | | Qwen | 22.33 | 28.68 | 14.44 | 22.55 | 9.33 | 12.16 | 31.33 | 42.14 | 19.56 | 26.54 |
> | DyPRAG-Combine | BM25 | 36.33 | 47.68 | 33.22 | 43.22 | 21 | 32.86 | 29.67 | 39.07 | 33.2 | 43.69 |
> | | all-MiniLM | 23.73 | 28.92 | 11.44 | 21.1 | 7.33 | 14.85 | 26.67 | 38.97 | 18.7 | 26.17 |
> | | Qwen | 24.27 | 29.95 | 10.89 | 20.58 | 8.33 | 18.92 | 28.33 | 39.43 | 19.07 | 26.98 |
>
>
> Our analysis indicates that while a stronger dense retriever generally **improves response quality and relevance** compared to weaker all-MiniLM, the BM25 algorithm used in our main paper **still outperforms the dense retriever** in this specific scenario, while also **maintaining significantly faster indexing speeds**. These new results further validate the rationale behind our original choice of BM25.
>
> ---
>
> With the discussion period ending soon, we wanted to check if our previous responses and the revised manuscript have satisfactorily addressed your comments.
>
> We have carefully considered your suggestions to improve the quality of our work. If you have any additional feedback or if there are specific points you feel we have not yet resolved, we would be grateful for the chance to discuss them further. Thank you sincerely for your constructive review.

---

### Official Review · Reviewer_kQ2T · 2025-10-31

**Soundness:** 2
**Presentation:** 3
**Contribution:** 1
**Rating:** 2
**Confidence:** 4

**Summary:**

This paper introduces DyPRAG, a novel framework that dynamically generates parametric knowledge adapters (LoRA modules) conditioned on retrieved documents during inference. Instead of statically training one LoRA per knowledge source as in PRAG, DyPRAG trains a parameter translator that maps document embeddings to LoRA weights, thereby enabling lightweight and flexible knowledge injection at runtime. The paper is well-organized, easy to follow, and clearly written, with a comprehensive set of experiments across multiple QA benchmarks.

**Strengths:**

1. The paper is well written and easy to understand. The motivation, method, and experimental setup are clearly explained, and figures are intuitive.

2. The idea of dynamically generating LoRA parameters from retrieved documents is fresh and interesting. It extends the RAG and PRAG paradigms in a meaningful way and shows the potential for a more adaptive integration between retrieval and parametric memory.

3. The authors evaluate DyPRAG across multiple QA datasets, ablation studies, and analyses, demonstrating solid empirical results.

**Weaknesses:**

1. It appears that DyPRAG requires training a parameter generator (translator) specifically for each backbone model (e.g., LLaMA, Mistral). This design makes it model-dependent, limiting its generality and practicality compared with standard RAG approaches, which can be easily applied to any model without retraining. A discussion or experiment on cross-model generalization (e.g., training the translator on one model and transferring to another) would make the work stronger

2. The experiments are conducted on relatively weak LLMs, which might exaggerate the benefits of parameter injection. For stronger models (e.g., Qwen3-8B/14B） that already have rich contextual understanding, explicit parameter injection might not be necessary. It would be valuable to see results on stronger models to test whether DyPRAG still brings improvements.

3. The paper mainly focuses on the parametric side but uses a single retriever setup. Since DyPRAG injects parameters conditioned on retrieved documents, errors in retrieval could directly amplify hallucinations, as incorrect LoRA weights may reinforce wrong knowledge. It would strengthen the work to include experiments with different retrievers (e.g., dense vs. hybrid, or advanced ones like Contriever, ColBERT, or Qwen retrievers) to analyze robustness to retrieval errors.

4. All metrics are reported as F1 scores, which can fluctuate significantly and are often sensitive to surface-level formatting differences (e.g., phrasing or extra explanations). F1 may yield high scores for semantically incorrect answers and thus is not fully reliable for evaluating generative models. I suggest adding LLM-as-a-judge evaluations (e.g., GPT-5-based scoring) as a complementary measure to verify consistency and correctness.

**Questions:**

The paper shows that DyPRAG-Combine achieves better performance than DyPRAG alone, but it is unclear why additional combination is needed if LoRA injection already encodes knowledge. Does this imply that the parametric generation is incomplete or that LoRA captures only partial information? A deeper analysis or explanation of why combining improves performance would be helpful.

---

> ### Author Response · Authors · 2025-11-20
>
> Dear reviewer kQ2T:
>
> Thank you for your sincere review and constructive feedback. We address your concerns as follows:
>
> > **W1: Model-dependent parameter translator.**
>
> We appreciate your observation. The coupling between a trained parameter translator and its corresponding backbone LLM is a common characteristic in deep learning and LLM research, as such components are typically designed to align with the architecture and parameter space of a specific model. The lack of cross-model generalization primarily arises from the task formulation itself: the parameter translator consumes intermediate representations and maps them into parameter spaces that vary across models in both shape and dimensionality.
>
> Nevertheless, our proposed DyPRAG provides substantial advantages that outweigh this non-transferability. It trains only a lightweight parameter translator at very low cost, yet still exhibits strong generalization in both IID and OOD settings. These benefits far exceed the limitations associated with model-specific coupling. Furthermore, in practical real-world deployments, a RAG system is generally built upon a single selected LLM. Thus, the dependency on a specific backbone does not limit the practical applicability of DyPRAG.
>
> > **W2: DyPRAG on up-to-date stronger models.**
>
> Thank you for raising this interesting concern. At present, stronger models commonly exhibit long-CoT behavior, which is characteristic of large reasoning models (LRMs). To evaluate these up-to-date stronger models, we selected **Qwen3-8B (a reasoning model)** and **Qwen3-4B-Instruct-2507 (an instruct model)** for our experiments. The complete analysis is provided in Appendix D, *Performance of DyPRAG on Non-Instruct Models* (Table 9, also shown below).
>
> The results indicate that the performance of Qwen3-8B drops substantially when applying DyPRAG-generated parameters. This decline is primarily attributable to differences in answer patterns: LRMs tend to produce very long reasoning chains, whereas our method augments only short and simple QA pairs. In contrast, the results for the recent instruct model, Qwen3-4B-Instruct, are consistent with our main findings, demonstrating that the current method is well-suited for instruct-style models.
>
> To enable compatibility with LRMs, **the parameter translation process would need to be integrated into the reinforcement learning training pipeline**. Addressing this challenge is an important direction for future work.
>
> | Base LLM | Method | 2WQA (Total) |  | HOA (Total) |  | POA |  | CWQ |  | Avg |  |
> |----------|--------|------|------|------|------|------|------|------|------|------|------|
> |  |  | EM | F1 | EM | F1 | EM | F1 | EM | F1 | EM | F1 |
> | Qwen3-8B | Vanilla | 24.07 | 31.33 | 21.00 | 28.12 | 0.00 | 0.40 | 22.33 | **36.01** | 17.00 | 23.97 |
> |  | RAG | **35.33** | **42.26** | **32.33** | **44.00** | **0.33** | **9.17** | **23.00** | 35.79 | **22.75** | **32.81** |
> |  | DyPRAG | 21.00 | 27.94 | 20.33 | 27.82 | 0.00 | 0.46 | 17.67 | 29.12 | 14.75 | 21.34 |
> |  | DyPRAG-Combine | 31.00 | 38.37 | 29.67 | 39.81 | **0.33** | 4.54 | 20.00 | 31.37 | 20.25 | 28.52 |
> | Qwen3-4B-Instruct | Vanilla | 21.00 | 28.97 | 15.00 | 23.32 | 8.67 | 12.10 | 0.00 | 1.58 | 11.17 | 16.49 |
> |  | RAG | 25.67 | 32.81 | 25.33 | 36.62 | 18.67 | 26.32 | 2.00 | 7.36 | 17.92 | 25.78 |
> |  | DyPRAG | 27.00 | 35.44 | 16.33 | 24.00 | 10.00 | 13.49 | 0.33 | 4.47 | 13.42 | 19.35 |
> |  | DyPRAG-Combine | **31.00** | **38.37** | **29.67** | **39.81** | **20.67** | **27.33** | **8.67** | **19.14** | **22.50** | **31.16** |
>
> > **W3: DyPRAG with different retrievers.**
>
> Thank you for raising this concern. Different retrievers can indeed affect the quality of retrieved documents. To evaluate the performance of dense retrieval, we adopt the well-known all-MiniLM-L6-v2 model as our dense retriever, which maps sentences into a 384-dimensional vector space. A complete analysis is provided in Appendix D, *Effect of Different Retrievers* (Table 13, also shown below). The results indicate that BM25 consistently outperforms simple dense retriever across all datasets under the DyPRAG framework. Across all settings, DyPRAG also consistently surpasses RAG, particularly in combination scenarios.
>
> Due to the extremely long indexing time required by dense retrievers, we postpone experiments with the stronger dense retrieval models you mentioned (e.g., Qwen3, ColBERT) to a later stage, where we will further evaluate their impact on both RAG and DyPRAG.

---

> ### Author Response · Authors · 2025-11-20
>
> | Method | Retriever | 2WQA | | HQA | | PQA | | CWQ | | Avg | |
> | :--- | :--- | :---: | :---: | :---: | :---: | :---: | :---: | :---: | :---: | :---: | :---: |
> | | | EM | F1 | EM | F1 | EM | F1 | EM | F1 | EM | F1 |
> | Vanilla | None | 30.00 | 36.43 | 19.89 | 28.64 | 4.67 | 7.96 | 30.00 | 42.44 | 24.43 | 31.85 |
> | RAG| Sparse | **28.40** | **34.20** | **19.13** | **28.67** | 5.67 | **16.13** | **25.33** | 35.45 | **23.04** | **30.86** |
> | | Dense | 22.20 | 27.82 | 11.66 | 20.60 | **6.33** | 12.15 | 24.67 | **36.48** | 17.70 | 24.95 |
> | DyPRAG| Sparse | **32.07** | **39.17** | **24.67** | **37.33** | **11.00** | **13.60** | **32.67** | **41.87** | **27.80** | **36.23** |
> | | Dense | 22.20 | 28.48 | 15.67 | 23.34 | 8.33 | 11.09 | 30.33 | 41.10 | 19.67 | 26.46 |
> | DyPRAG-Combine | Sparse | **36.33** | **47.68** | **33.22** | **43.22** | **21.00** | **32.86** | **29.67** | **39.07** | **33.20** | **43.69** |
> | | Dense | 23.73 | 28.92 | 11.44 | 21.10 | 7.33 | 14.85 | 26.67 | 38.97 | 18.70 | 26.17 |
>
> > **W4: Limitation of F1 scores.**
>
> Thank you for your thoughtful comments. As you noted, token-level F1 scores may produce high scores even for semantically incorrect answers. We therefore evaluated an LLM-as-a-judge approach using GPT-4o and found that its judgments are consistent with the widely used and more efficient **answer-level Exact Match (EM) metric** in RAG systems [1, 2, 3]. Based on this finding, we have updated our manuscript to include EM scores in the main experiments, providing a stricter evaluation of generated text. As shown in Table 1, our method still achieves the best performance under the EM metric, further demonstrating the effectiveness of DyPRAG under both F1 and EM evaluations.
>
> | LLaMA3.2-1B | 2WQA | HQA | PQA | CWQ | AVG |
> |-------------|------|------|------|------|------|
> | Vanilla | 18.00 | 19.00 | 9.00 | 32.33 | 19.58 |
> | RAG | 21.33 | 34.00 | 16.67 | 35.67 | 26.92 |
> | PRAG | 19.33 | 19.33 | 24.00 | 33.67 | 24.08 |
> | PRAG-Combine | 23.33 | 31.33 | 52.33 | 40.00 | 36.75 |
> | DyPRAG | 21.33 | 21.33 | 12.67 | 35.33 | 22.67 |
> | DyPRAG-Combine | **25.00** | **34.33** | **57.00** | **43.33** | **39.92** |
>
> > **Q1: A deeper analysis or explanation of why combining improves performance.**
>
> Thank you for raising this concern. It is reasonable to question why DyPRAG may not always be sufficient on its own. In the IID experiments, where all datasets involve commonsense QA and documents are retrieved from Wikipedia using BM25, DyPRAG consistently outperforms RAG across all model sizes, though it occasionally underperforms PRAG. This is understandable: PRAG is explicitly fine-tuned for each document, whereas DyPRAG’s strength lies in dynamically transforming documents into LoRA parameters. Given that this process involves encoding text into embeddings and subsequently translating embeddings into parameters, some degree of information loss is expected.
>
> In the OOD experiments discussed in Section 4.4, where all datasets are equipped with golden passages, the vanilla model performs poorly due to a lack of sufficient relevant knowledge, especially in the IIRC dataset. On the other hand, RAG shows strong performance. This demonstrates that many tasks require fine-grained information from documents to answer questions accurately. As a result, **we acknowledge that DyPRAG has limitations in performing fine-grained knowledge transformation. This limitation arises naturally from the challenges of converting information across multiple modalities.** However, even though DyPRAG focuses on coarse-grained knowledge transformation, this process enhances the overlap between different types of knowledge, improving the model’s capability to understand unseen documents and showing best performance in both IID and OOD.
>
> A detailed discussion is provided in Section 4.6, *Pre-inject Converted Parameters Enhance Knowledge Overlap* (Table 5). Our analysis shows that pre-injecting document knowledge reduces uncertainties caused by the low overlap between internal and external knowledge. Therefore, we argue that the key contribution of DyPRAG does not lie in achieving lossless cross-modal knowledge conversion. **Instead, its value comes from dynamically optimizing the model’s internal knowledge representation for
> each question, ultimately enhancing the traditional RAG framework to produce better responses.**
>
> **We have revised and improved our manuscript accordingly, adding more experiments in both the main text and the appendix. We sincerely welcome your further comments!**
>
> [1] Su, W., et al. "Dragin: Dynamic retrieval augmented generation based on the real-time information needs of large language models. ACL2024.
>
> [2] Jiang, Zhengbao, et al. "Active retrieval augmented generation." EMNLP2023.
>
> [3] Ram, Ori, et al. "In-context retrieval-augmented language models." TACL2023.

---

> ### Author Response · Authors · 2025-11-26
>
> Dear Reviewer kQ2T,
>
> To thoroughly evaluate **the impact of different retrievers as suggested in W3**, we have conducted additional experiments using the stronger **Qwen3-Embedding-0.6B model (1024-dimensional)** as our dense retriever.
>
> | Method | Retriever | 2WQA |  | HQA |  | PQA |  | CWQ |  | Avg |  |
> |--------|-----------|------|-------|-------|-------|------|-------|-------|-------|-------|-------|
> | | | EM | F1 | EM | F1 | EM | F1 | EM | F1 | EM | F1 |
> | Vanilla | None | 30 | 36.43 | 19.89 | 28.64 | 4.67 | 7.96 | 30 | 42.44 | 24.43 | 31.85 |
> | RAG | BM25 | 28.4 | 34.2 | 19.13 | 28.67 | 5.67 | 16.13 | 25.33 | 35.45 | 23.04 | 30.86 |
> | | all-MiniLM | 22.2 | 27.82 | 11.66 | 20.6 | 6.33 | 12.15 | 24.67 | 36.48 | 17.7 | 24.95 |
> | | Qwen | 19.53 | 25.86 | 12.11 | 19.57 | 6.63 | 13.79 | 25.67 | 36.36 | 16.63 | 23.82 |
> | DyPRAG | BM25 | 32.07 | 39.17 | 24.67 | 37.33 | 11 | 13.6 | 32.67 | 41.87 | 27.8 | 36.23 |
> | | all-MiniLM | 22.2 | 28.48 | 15.67 | 23.34 | 8.33 | 11.09 | 30.33 | 41.1 | 19.67 | 26.46 |
> | | Qwen | 22.33 | 28.68 | 14.44 | 22.55 | 9.33 | 12.16 | 31.33 | 42.14 | 19.56 | 26.54 |
> | DyPRAG-Combine | BM25 | 36.33 | 47.68 | 33.22 | 43.22 | 21 | 32.86 | 29.67 | 39.07 | 33.2 | 43.69 |
> | | all-MiniLM | 23.73 | 28.92 | 11.44 | 21.1 | 7.33 | 14.85 | 26.67 | 38.97 | 18.7 | 26.17 |
> | | Qwen | 24.27 | 29.95 | 10.89 | 20.58 | 8.33 | 18.92 | 28.33 | 39.43 | 19.07 | 26.98 |
>
>
> Our analysis indicates that while a stronger dense retriever generally **improves response quality and relevance** compared to weaker all-MiniLM, the BM25 algorithm used in our main paper **still outperforms the dense retriever** in this specific scenario, while also **maintaining significantly faster indexing speeds**. These new results further validate the rationale behind our original choice of BM25.
>
> ---
>
> With the discussion period ending soon, we wanted to check if our previous responses and the revised manuscript have satisfactorily addressed your comments.
>
> We have carefully considered your suggestions to improve the quality of our work. If you have any additional feedback or if there are specific points you feel we have not yet resolved, we would be grateful for the chance to discuss them further. Thank you sincerely for your constructive review.

---

### Official Review · Reviewer_WGwD · 2025-11-01

**Soundness:** 3
**Presentation:** 3
**Contribution:** 2
**Rating:** 4
**Confidence:** 4

**Summary:**

The paper introduces Dynamic Parametric RAG (DyPRAG), a retrieval-augmented generation framework that dynamically converts retrieved documents into lightweight LoRA adapters during inference, eliminating the need for storing per-document parameters like in PRAG or handling long contexts as in RAG. A parameter translator learns to map document embeddings to LoRA weights, enabling on-the-fly parameter generation for unseen documents. This design significantly reduces storage and computation costs while improving retrieval-based reasoning and mitigating knowledge conflicts. The authors also propose DyPRAG-Combine, which fuses dynamic parameter injection with contextual retrieval for enhanced factuality and consistency. Experiments across QA benchmarks demonstrate DyPRAG’s superior efficiency, generalization, and performance compared to both PRAG and traditional RAG methods.

**Strengths:**

1) The paper targets the high training and inference cost of parameter-injection RAG methods and proposes a learnable neural network (Translator) that generates LoRA parameters from documents, writing document knowledge into LoRA.

2) DyPRAG-Combine achieves the best performance and improves over RAG, suggesting that LoRA generated and injected by the Translator helps alleviate conflicts between parametric knowledge and external evidence.

3) The experimental comparisons are relatively comprehensive; without per-document LoRA training, the method still maintains a small gap from PRAG.

**Weaknesses:**

1) Using only the “last token’s final-layer hidden state” as the document vector may limit representation capacity.

2) The Translator architecture is relatively simple, and there is a lack of systematic analysis on its model size and parameter configuration.

3) The paper provides only a partial and qualitative analysis of inference latency; end-to-end latency measurements are missing.

**Questions:**

1) When documents are long or key information is sparse, can the last-token hidden state still effectively represent the whole document?

2) In Table 1, DyPRAG outperforms RAG, but in the OOD results of Table 2, DyPRAG is clearly worse than RAG. What causes this
discrepancy? Does it indicate insufficient generalization of the Translator’s document-to-parameter mapping?

---

> ### Author Response · Authors · 2025-11-20
>
> Dear reviewer WGwD:
>
> Thank you for your sincere review and constructive feedback. We address your concerns as follows:
>
> > **W1: Using only the “last token’s final-layer hidden state” may limit representation capacity.**
>
> Thank you for raising this insightful point. Given the auto-regressive nature of the Transformer decoder, the final-layer hidden state of the last token theoretically encapsulates the full context of the input document. While it is possible to use a bi-directional sentence encoder to obtain embeddings, we argue that introducing an external encoder creates a representational gap. Bridging this gap, learning the transformation from the external embedding to our base model parameter space, requires significant data (e.g., previous work [1] has necessitated training on over 800,000 examples to achieve this).
>
> This contradicts our design philosophy, as we prioritize a method that is simple, direct, and efficient. Our experiments in both IID and OOD settings demonstrate that the last token’s hidden state contains sufficiently rich information, ensuring that the mapping from embedding to LoRA parameters yields robust performance across all scenarios.
>
> > **W2: The Translator architecture is relatively simple and lack of systematic analysis on its model size and parameter configuration.**
>
> Thank you for your thoughtful comments. A hypernetwork is a neural network that generates the parameters of another base network, typically with a much smaller number of parameters. Recent studies have generally adopted a simple MLP as the hypernetwork architecture [1, 2, 3, 4, 5]. These works, together with our experiments, demonstrate that such a relatively lightweight hypernetwork is sufficient to achieve strong performance. Indeed, DyPRAG exhibits substantial improvements and strong generalization ability.
>
> In the previous version of our manuscript, we analyzed the effect of parameter configurations within the parameter translator in Appendix D due to page limitations. Following your valuable suggestion, we have now added a brief introductory paragraph in Section 4.5 that directs readers to the detailed analysis provided in Appendix D, *Effect of Parameter Translator Size*  (Table 11). The results indicate that even an extremely small hypernetwork is capable of learning the underlying transformation from the document representation to the corresponding parameters.
>
> > **W3: End-to-end latency measurements are missing.**
>
> Thank you for raising this concern. In the previous version, we analyzed the end-to-end latency as the number of injected documents increased, and reported the results in the Appendix. Following your suggestion, we now include a short concluding remark at the end of Section 4.5, *Effect of Parameter Translator Size* that points readers to the detailed analysis in Appendix D, *Computation Effect of Injected Documents Number* (Table 12, also presents in next comment).
>
> Our findings show that DyPRAG achieves superior inference efficiency compared to standard RAG, particularly as the number of injected documents increases. While the inference time of standard RAG grows substantially with more injected documents, DyPRAG-Combine consistently maintains lower inference latency, primarily due to its shorter response lengths (Figure 4). Moreover, when passages are excluded from the input context, DyPRAG still significantly outperforms standard RAG in inference time.
>
> However, the design of DyPRAG introduces additional encoding and translation overhead as the number of injected documents increases. At present, these encoding and translation procedures are not fully optimized. In real-world deployments, RAG-based queries are typically managed through message queueing systems (e.g., Kafka), which naturally support asynchronous execution. Under such an architecture, document embeddings can be computed and translated during the waiting period using separate processes or model instances, effectively mitigating this overhead.
>
>
> [1] Cheng, Xin, et al. "xrag: Extreme context compression for retrieval-augmented generation with one token." NIPS2024.
>
> [2] Chen, Tong, et al. "Generative adapter: Contextualizing language models in parameters with a single forward pass." ICLR2025.
>
> [3] Charakorn, Rujikorn, et al. "Text-to-LoRA: Instant Transformer Adaption." ICML2025.
>
> [4] Liao, Huanxuan, et al. "From instance training to instruction learning: Task adapters generation from instructions." NIPS2024.
>
> [5] Liao, Huanxuan, et al. "Awakening augmented generation: Learning to awaken internal knowledge of large language models for question answering." COLING2025.

---

> ### Author Response · Authors · 2025-11-20
>
> | Documents | Method          | Inference Time | Loading Time | Translate Time | Encode Time |
> |-----------|-----------------|----------------|--------------|----------------|-------------|
> | 3         | DyPRAG          | 0.84           | 0.0037       | 0.056          | 0.132       |
> |           | RAG             | 1.23           | -            | -              | -           |
> |           | DyPRAG-Combine  | 0.36           | 0.0037       | 0.055          | 0.132       |
> | 10        | DyPRAG          | 0.80           | 0.0044       | 0.185          | 0.433       |
> |           | RAG             | 1.54           | -            | -              | -           |
> |           | DyPRAG-Combine  | 0.78           | 0.0045       | 0.185          | 0.432       |
> | 20        | DyPRAG          | 0.80           | 0.0057       | 0.361          | 0.862       |
> |           | RAG             | 1.74           | -            | -              | -           |
> |           | DyPRAG-Combine  | 1.40           | 0.0057       | 0.361          | 0.862       |
> | 30        | DyPRAG          | 0.80           | 0.0067       | 0.545          | 1.295       |
> |           | RAG             | 2.18           | -            | -              | -           |
> |           | DyPRAG-Combine  | 1.96           | 0.0067       | 0.545          | 1.294       |
>
> > **Q2: What and Why causes this discrepancy in DyPRAG in OOD?**
>
> Thank you for raising this concern. It is reasonable to question why DyPRAG may not always be sufficient on its own. In the IID experiments, where all datasets involve commonsense QA and documents are retrieved from Wikipedia using BM25, DyPRAG consistently outperforms RAG across all model sizes, though it occasionally underperforms PRAG. This is understandable: PRAG is explicitly fine-tuned for each document, whereas DyPRAG’s strength lies in dynamically transforming documents into LoRA parameters. Given that this process involves encoding text into embeddings and subsequently translating embeddings into parameters, some degree of information loss is expected.
>
> In the OOD experiments (Section 4.4), where all datasets are provided with golden passages, the vanilla model performs poorly because it lacks sufficient task-specific knowledge—particularly on the IIRC dataset. RAG, by contrast, performs strongly, indicating that many tasks require fine-grained document information to answer questions accurately. Consequently, we attribute **the performance discrepancy in the OOD setting to DyPRAG’s limitations in capturing fine-grained knowledge transformations rather than to insufficient generalization.** Notably, DyPRAG outperforms the vanilla model across all OOD datasets, demonstrating effective parametric knowledge enhancement.
>
> **This limitation is expected given the inherent difficulty of converting information across modalities.** Nevertheless, despite focusing on coarse-grained knowledge transformation, DyPRAG increases the overlap between internal and external knowledge representations, thereby improving the model’s ability to interpret unseen documents. This leads to the best performance in the combined IID and OOD setting.
>
> A detailed discussion is provided in Section 4.6, *Pre-inject Converted Parameters Enhance Knowledge Overlap* (Table 5). Our analysis shows that pre-injecting document knowledge reduces uncertainties caused by the low overlap between internal and external knowledge. **Therefore, we argue that the key contribution of DyPRAG does not lie in achieving lossless cross-modal knowledge conversion. Instead, its value comes from dynamically optimizing the model’s internal knowledge representation for each question, ultimately enhancing the traditional RAG framework to produce better responses.**
>
> > **Q1: When documents are long or key information is sparse, can the last-token hidden state still effectively represent the whole document?**
>
> Thank you for raising this concern. As discussed in Q2, DyPRAG is not designed to reconstruct all fine-grained information from documents. Its strength is helping the model interpret retrieved documents through parametric transformation. This process naturally loses some information because it moves across three representations: text, embeddings and parameters.
>
> In the IID experiments, where ground truth passages are not available, the retriever often returns documents with limited key information. Even in these cases, DyPRAG can extract useful parametric cues and improve model performance. Therefore, when documents lack essential details or are too long for the last-token hidden state to capture fine-grained content, DyPRAG can still distill parameter-level knowledge that helps the model internalize the document and generate better answers. This will be an interesting future work that DyPRAG can further improve on it.
>
> **We have revised and improved our manuscript accordingly, adding more experiments in both the main text and the appendix. We sincerely welcome your further comments!**

---

> ### Author Response · Authors · 2025-11-26
>
> Dear Reviewer WGwD,
>
> With the discussion period ending soon, we wanted to check if our previous responses and the revised manuscript have satisfactorily addressed your comments.
>
> We have carefully considered your suggestions to improve the quality of our work. If you have any additional feedback or if there are specific points you feel we have not yet resolved, we would be grateful for the chance to discuss them further. Thank you sincerely for your constructive review.

---

### Official Review · Reviewer_cWye · 2025-11-01

**Soundness:** 2
**Presentation:** 3
**Contribution:** 2
**Rating:** 4
**Confidence:** 2

**Summary:**

The paper proposes DyPRAG to solve the problems of efficiency of PRAG (Parametric RAG). Instead of generating a set of versions for a document and parametrize them at inference time in PRAG, DyPRAG trains a small hypernetwork to encode a documentfrom its original embedding (last layer of transformer). It uses the same process to generate training data as in PRAG. The hypernetwork is trained to match the document representation in the training set. By doing so, DyPRAG does not need to store the encoded document embedding but can generate it efficiently. It is claimed that the training time is also reduced.
The experiments show that DyPRAG can lead to effectiveness close to PRAG, and outperforms other RAG approaches.

**Strengths:**

1. The paper targets an important problem in PRAG - that of efficiency. It proposes an efficient solution to create document representation online, thus save the storage space. The proposed method encodes a document by a trained parameter translator based on the same training data as PRAG. The time complexity is comparable. What is faster is the online inference time: document representation can be obtained through the parameter translator, without the costly generation of multiple versions and their encoding.

2. The experimental results show that DyPRAG can achieve a performance close to PRAG in some cases.

3. The paper contains many analyses, showing the impact of different components in the training loss function, the complexity, generalizability, and the impact of different hyperparameters. The analyses are informative.

**Weaknesses:**

1. As DyPRAG does not need to generate multiple versions for a document at inference time, but generates a representation directly from the encoded embedding and the parameter translator, it is unclear if the created representation can benefit from the implicit expansion effect in PRAG through the multiple versions of document. This aspect has not been analyzed. This cannot be seen directly from the global effectiveness measures. that are quite variable.

2. One weakness of PRAG is that the document representation is created independently of the query. It is uncertain that the information relevant to the query is always included in the representation. DyPRAG inherits the same weakness. It would be better to consider the query when creating document representation.

3. The experimental results are not consistent when comparing DyPRAG with others (namely PRAG). DyPRAG often needs to be combined with in-context injection to be competitive. However, the combination will reduce its efficiency, making it a less interesting solution to RAG.

**Questions:**

1. Have you analyzed the quality of the generated document representation, in comparison to PRAG? The analysis would be interesting to understand if the creation of multiple versions of a document is necessary. If representations of similar quality can be obtained in DyPRAG and PRAG, then there is indeed no need to create multiple versions of the document, and this can be done through parameter translator.

---

> ### Author Response · Authors · 2025-11-20
>
> Dear reviewer cWye:
>
> Thank you for your thorough review and constructive feedback. We address your concerns as follows:
>
> > **W1: Implicit expansion effect of the multiple versions of document.**
>
> Thank you for this valuable suggestion. During the offline collection of Doc-Param pairs, we applied data augmentation to the original passages to enable the LoRA parameters to better encode knowledge. An additional ablation study is provided in Section 4.5, with the corresponding table included below. The results indicate that data augmentation plays a critical role in parametrization for better knowledge memorization and manipulation. Removing data augmentation significantly diminishes the parametric knowledge, thereby hindering DyPRAG’s ability to learn the transformation of knowledge from documents to parameters.
>
> | Method         | 2WQA (Total) | HQA (Total) | PQA   | CWQ   | HIRC  | SQA   | OBQA  | MQA   |
> |----------------|--------------|-------------|-------|-------|-------|-------|-------|-------|
> | Vanilla        | 26.87        | 17.76       | 2.87  | 26.47 | 8.78  | 1.00  | 40.09 | 33.67 |
> | RAG            | 24.31        | 20.73       | 9.97  | 28.23 | 30.52 | 39.00 | 45.00 | 52.67 |
> | PRAG-Combine   | 27.49        | 23.10       | 23.43 | 32.13 | —     | —     | —     | —     |
> | w/o Aug        | 22.79        | 19.00       | 10.74 | 28.54 | —     | —     | —     | —     |
> | *Change*       | -17.1%        | -17.7%      | -54.2% | -11.2%| —     | —     | —     | —     |
> | DyPRAG         | 26.46        | 19.67       | 6.64  | 31.94 | 10.23 | 15.67 | 43.38 | 34.67 |
> | w/o Aug        | 28.36        | 15.71       | 3.35  | 28.04 | 8.49  | 0.30  | 38.36 | 22.94 |
> |*Change*         | +7.2%        | -20.1%       | -49.5% | -12.2%| -17.0% | -98.1% | -11.6%| -33.8% |
> | DyPRAG-Combine | 25.18        | 27.57       | 22.69 | 33.57 | 38.25 | 43.33 | 48.57 | 52.67 |
> | w/o Aug        | 23.00        | 19.88       | 9.84  | 27.97 | 29.41 | 30.67 | 43.90 | 34.03 |
> |*Change*         | -8.7%        | -27.9%       | -56.6% | -16.7%| -23.1%| -29.2% | -9.6% | -35.4% |
>
> > **W2: Consider the query when creating document representation**
>
> Thank you for pointing this out. It is indeed true that we are uncertain whether the information relevant to the query is consistently included in the representation, as this is highly dependent on the retriever. Following your suggestion, we transformed the query+document into parametric knowledge. However, the results showed no consistent improvement when incorporating the query during the generation of representations.
>
> We believe this is because a query is a question, not a illustration of the factual knowledge required to answer it. Directly injecting the query does not enhance the model's parametric knowledge. Therefore, for PRAG and DyPRAG, which follow the RAG paradigm, improving how the query retrieves more relevant documents will likely be more beneficial for generating better document representations.
>
> | Method | 2WQA|  | HotpotQA |  | PQA |  | CWQ |  |
> |--------|---------|---------|----------|---------|---------|---------|---------|---------|
> | | EM | F1 | EM | F1 | EM | F1 | EM | F1 |
> | Vanilla | 21.00 | 26.87 | 9.60 | 17.76 | 0.67 | 2.87 | 18.00 | 26.47 |
> | RAG | 17.00 | 24.21 | 11.33 | 20.73 | 0.67 | 9.97 | 18.64 | 28.22 |
> | DyPRAG | 20.00 | 26.46 | 11.33 | 19.67 | 3.00 | **6.64** | **22.67** | **31.94** |
> | DyPRAG+Query | **20.33** | **26.51** | 11.33 | **19.80** | 3.00 | 6.51 | 22.33 | 31.61 |
> | DyPRAG-Combine | 19.00 | 25.18 | 18.67 | **27.57** | 7.33 | 22.69 | **23.67** | **33.57** |
> | DyPRAG-Combine+Query | **19.33** | **25.55** | 18.67 | 27.49 | 7.33 | **22.87** | 23.33 | 33.13 |

---

> ### Author Response · Authors · 2025-11-20
>
> > **W3.1:  Sometime DyPRAG needs to be combined with in-context injection to be competitive.**
>
> Thank you for raising this concern. It is natural to question why DyPRAG alone might not be sufficient all the time. In IID experiments, where all datasets are based on commonsense QA and documents are retrieved from Wikipedia using BM25, DyPRAG consistently achieves better performance compared to RAG across all models, although it occasionally underperforms PRAG. This is understandable since PRAG is specifically fine-tuned for each document, while DyPRAG’s key strength lies in its ability to dynamically transform documents into LoRA parameters. It is reasonable to assume that the process of encoding and translating, which involves converting text into embeddings and then embeddings into parameters, may lead to some degree of information loss.
>
> In the OOD experiments discussed in Section 4.4, where all datasets are equipped with golden passages, the vanilla model performs poorly due to a lack of sufficient relevant knowledge, especially in the IIRC dataset. On the other hand, RAG shows strong performance. This demonstrates that many tasks require fine-grained information from documents to answer questions accurately. As a result, we acknowledge that DyPRAG has limitations in performing fine-grained knowledge transformation. **This limitation arises naturally from the challenges of converting information across multiple modalities.** However, even though DyPRAG focuses on coarse-grained knowledge transformation, this process enhances the overlap between different types of knowledge, improving the model’s capability to understand unseen documents and showing best performance in both IID and OOD.
>
> A detailed analysis can be found in Section 4.6, *Paragraph Pre-inject Converted Parameters Enhances Knowledge Overlap* (Table 5). We observed that pre-injecting document knowledge into the model helps reduce uncertainties caused by the low overlap between internal and external knowledge. **Therefore, we believe that the core contribution of DyPRAG is not in achieving lossless knowledge conversion across modalities. Instead, its value lies in dynamically optimizing the model’s internal representation of knowledge for a given question, building upon the traditional RAG framework to deliver better responses.**
>
> > **W3.2: The combination will reduce its efficiency.**
>
> Thank you for your concern. We provide a detailed end-to-end latency analysis in Appendix D, *Computation Effect of Injected Documents Number* (Table 12). In practice, this combination does not impact the efficiency of the model. By injecting parameterized knowledge through DyPRAG, the model significantly enhances its knowledge internalization, resulting in a much shorter generation length compared to RAG (Figure 4). This reduction in response length greatly decreases inference time.
> The remaining encoding and translation time can be optimized using message queues (e.g., Kafka), which naturally support asynchronous execution environments in real-world RAG applications. **Therefore, this combination not only improves the quality of responses and the model's understanding of knowledge but also ensures high efficiency.**
>
> > **Q1: The quality of the generated document representation between PRAG and DyPRAG.**
>
> Thank you for raising this question. One way to evaluate the quality of the generated document representations between PRAG and DyPRAG is through performance comparison. As shown in Table 1, when using parameters alone, only LLaMA3-1B achieves better performance. However, under all model settings, our combination achieves superior performance, even in OOD scenarios. This suggests that the document representations generated by DyPRAG align better with the given context and provide more useful knowledge for inference.
>
> We also analyzed the similarity between knowledge generated by DyPRAG and PRAG using the same passages. Our findings indicate a high similarity in their parameter representations (>75%), though there are some differences due to the variations in training methods. Overall, the parameter representations generated by DyPRAG are of sufficiently high quality and exhibit strong similarity to those of PRAG, making it entirely feasible to replace PRAG with the parameter translator in DyPRAG.
>
> | 2WQA | HQA | PQA | CWQ |
> |------|-----|-----|-----|
> | 78.85 | 78.55 | 76.05 | 78.47 |
>
> **We have revised and improved our manuscript accordingly, adding more experiments in both the main text and the appendix. We sincerely welcome your further comments!**

---

> ### Author Response · Authors · 2025-11-26
>
> Dear Reviewer cWye,
>
> With the discussion period ending soon, we wanted to check if our previous responses and the revised manuscript have satisfactorily addressed your comments.
>
> We have carefully considered your suggestions to improve the quality of our work. If you have any additional feedback or if there are specific points you feel we have not yet resolved, we would be grateful for the chance to discuss them further. Thank you sincerely for your constructive review.

---

### Note · Authors · 2026-01-05

I have read and agree with the venue's withdrawal policy on behalf of myself and my co-authors.